# Use of Modern Technologies for the Conservation of Historical Heritage in Water Management

**Adrian Șmuleac** [1], **Laura Șmuleac** [1,*] , **Teodor Eugen Man** [2], **Cosmin Alin Popescu** [1], **Florin Imbrea** [1], **Isidora Radulov** [1], **Tabita Adamov** [1] **and Raul Pașcalău** [1,*]

[1]  Faculty of Agriculture, Banat's University of Agricultural Sciences and Veterinary Medicine "King Michael I of Romania" from Timisoara, 300645 Timisoara, Romania; adriansmuleac.as@gmail.com or adrian_smuleac@usab-tm.ro (A.Ș.); cosminpopescu@usab-tm.ro (C.A.P.); florinimbrea@usab-tm.ro (F.I.); isidoraradulov@usab-tm.ro (I.R.); tabitadamov@usab-tm.ro (T.A.)

[2]  Hydrotehnical Department, Politehnica University Timisoara, 300022 Timisoara, Romania; eugen@zavoi.ro

\*  Correspondence: laurasmuleac@usab-tm.ro (L.Ș.); raulpascalau@yahoo.com (R.P.)

**Abstract:** Historical monuments represent a cultural heritage that humanity has a duty to preserve and conserve. Lately all over the world, scanning these heritage objectives has become a priority, in order to preserve in the smallest details the used architecture. The work aims to complete the cultural heritage for Sânmihaiu Român hydro technical development built between 1912 and 1915, located on the Bega River in Western Romania, through modern mobile scanning technology, Leica Pegasus Backpack, necessary for the creation of a three-dimensional (3D) documentation, for the completion of the cultural heritage, and for the creation of a 3D database. The purpose of the scientific paper is restoring Sanmihaiu Roman Hidro technical Node, subject to degradation, in order to achieve the project "The navigable Bega", waterway connection to Serbia. Collecting method of LiDAR data is Fused Slam, the acquisition of RINNEX data being made by placing a Leica GS08 Master Station. Visualization of quality graphics has been performed in Quality Control (QC) Tools. The scanning accuracy is between 2 and 3 cm and the 3D data processing was performed with the Cyclone Model version program, with SmartPick Point and Virtual Surveyor functions. The obtained point clouds will be of a great help in order to follow in time the construction which can be used whenever it will be needed by the designers and specialists in the field of hydrotechnics.

**Keywords:** lock; water; Leica Pegasus; radmin; zero velocity update algorithm (ZUPT); inertial navigation systems (INS); inertial measurement unit (IMU); 3D; simultaneous location and mapping (SLAM); Mobile Mapping System (MMS)

## 1. Introduction

The evolution of technologies in the last half of the 21st century has led to huge advances in electronics, IT, graphics (digital three-dimensional (3D) models), making it possible to develop laser, terrestrial (TLS - Terrestrial Laser Scanning and MMS—Mobile Mapping System), and aerial (UAV Unmanned aerial vehicle) scanning technology. Thus, the possibility of processing dense points clouds in an efficient and cost-effective manner has facilitated a multitude of applications concerning the acquisition of 3D data in areas such as, topography, environmental protection, water management [1,2], the completion and preservation of cultural heritage, forest resources, etc. [3,4].

Due to the evolution of techniques and equipment at an accelerated pace, the creation of virtual reality is no longer a major problem. The software dedicated to this purpose and used in this work, in addition to the modern equipment used [5], was the key to the success achieved. In addition to virtual reality, we can talk today about "Virtual Surveying", or a virtual tour and the creation of a viewing

link (TruView), is no longer a problem, and the obtained data can provide a broad perspective on the scanned lens. A single 3D scan can be used to carry out several projects (design, general cadaster, civil engineering, water supply network, gas network, GIS—Geographic Information System, forestry, and topography).

As the world's population grows and global changes in construction and infrastructure become faster [5,6], the need to document this growth and change increases. Referring to image capture sensor systems, this new concept provides us data regarding the way we can shape the surrounding environment [7–9]. This new technology has applicability in many areas or specializations, including survey (rail, infrastructure, and utility), industry, GIS, hydrography, geodesic measurements [10], 3D laser scanning [11], monitoring, and mobile mapping. This mobile mapping solution is designed for rapid and regular capture of reality. It is fully portable, allowing it to be checked as luggage if you want to take a flight. With the development of laser scanning technology for the topographic side, it can be used as easily as possible in order to get results which were impossible some time ago. Measuring lands and buildings can be done taking advantage of short time in which thousands to millions of points can be measured without the need for them to be marked. The advantage of laser scanning technology is that it is possible to measure a construction, if access to it is impossible (due to the constructive system or if it is affected by different calamities) to perform measurements using classical methods [12,13].

In 2013, Corso and Zakhor [14] used an arrangement of five LASER (light amplification by stimulated emission of radiation) scanners in a platform with backpacks combined with two cameras and an inertial measurement unit (IMU), aiming internal mapping. The authors obtained a centimeter precision in the 3D reconstruction. Another backpack platform using a mobile LASER scanning system for indoor and outdoor environments was evaluated in 2015 by Lauterbach et al. [15]. These authors developed a free global navigation satellite systems (GNSS) approach based on LASER scanners (SICK LMS 100 and Riegl VZ-400) and a low-cost IMU (Phidgets 1044) for simultaneous location and mapping solution (SLAM). A GNSS-free approach using an extended Kalman Filter (EKF) method for indoor environments was also discussed by Wen and colleagues in 2016 [16]. The system proposed by them has three 2-D LASER scanners (UTM-30LX) and an IMU (Xsens MTi-10) specially arranged in a backpack. Commercial solutions with a LASER scanner and omnidirectional systems have recently been introduced by Leica (Leica Geosystems Leica Pegasus: Backpack) [17] and Google [18].

The mobile mapping system is a high-precision 3D mobile measurement system, capable of efficiently obtaining accurate 3D position data, for buildings, roads, traffic signs, more precisely everything that we see with our eyes, at a distance of maximum 70 m left–right, with a 360 degree view [19,20].

Very interesting examples of MMS comparisons are reported in Thomson [21–23]. In 2013, Thomson and colleagues [24] investigated two very different products: the Viametris i-MMS stroller without GNSS/IMU sensors and the portable 3D Laser Mapping/CSIRO ZEB1 device, while performing internal scans purchased using a Faro Focus 3D [25] as a reference. Nocerino and collaborators, in 2017, compare two similar systems, GeoSlam Zeb-Revo [26] portable and Leica Pegasus Backpack, the latter equipped with a GNSS receiver. One of the tests was performed in a two-story building, scanned with a Leica HDS7000, and the other in a large square (80 m × 70 m), previously scanned with a "classic" MMS, namely with a Riegl VMX -450 on a van. The comparison made by Lehtola and collaborators in 2017 [21], is quite impressive, as it makes a clear comparison between no less than eight different systems: five commercial cameras—the innovative 3D camera Matterport, the NavVis stroller, Zebedee (the oldest Zeb model), the rear hand handle, the Kaarta Stencil handle and again the Leica Pegasus backpack-and three interesting research prototypes—the Aalto VILMA rotating wheel, the FGI (Finnish Geospatial Research Institute) Slammer, and the Würzburg backpack.

Of course, MMS (mobile mapping system) models can be compared with models from SfM-based photogrammetric topography, while keeping statically acquired TLS (Terrestrial Laser Scanning) data as a reference [7].

## 2. History and Importance of Sânmihaiu Român Hydro Technical Node from a Hydrological and Cultural Point of View

The hydrotechnical node at Sânmihaiul Român is located upstream of Sânmihaiu Român, being presently cultural heritage, about 1 km from the Sânmihai bridge over the Bega canal, next to km 28 + 200 of the left bank pier, measured from the border with Serbia.

The construction of the entire Hydrotechnical Node was carried out between 1912 and 1915. The hydrotechnical node consists of the dam and the mechanism house, a lock and the exploitation canton. Navigation on the canal in the Romanian sector dates back to 1860, which was closed due to the political situation in 1958, a situation that continues today. On the Serbian territory, the navigable sector of the Bega waterway connects with the Danube-Tisza-Danube hydro-amelioration and navigation system [27,28]. For this reason, the Bega canal has a strategic position and makes possible the connection between the western part of Romania with the North Sea and the Black Sea through the Rhine-Main-Danube canal [29].

In the period 1716–1778 (according to historical writings), Timişoara, formally Banat, was under imperial administration. The priority at that time was the drying up of the swamps, for the economic development of the Banat province, which made the air unhealthy and the water unbearable. They also sought to find optimal solutions for transporting wood and agricultural products. In order to achieve and achieve these objectives, it was necessary channeling, at that time, the Bega River. The first to start this project was General Florimund Mercy, also appointed governor of Banat Timisoara, on the recommendation of Prince Eugene of Savoy. Mercy implemented the plan to drain the mud in 1728 by regularizing the watercourse and channeling the Bega River.

The construction of the canal started near Făget, near Timişoara, from where four smaller canals were opened, with locks, in the direction of the Fabric suburb, to supply the water mills, to serve the factories, and to ensure the navigation in order to transport the firewood, the wood for construction and the salt. The Bega River thus became navigable, and the city of Timisoara was supplied with good quality drinking water. At the end of 1732, the first ship was launched to Pancevo, where due to the numerous sand thresholds, the route was abandoned.

However, between 1735 and 1754, a new canal variant was built, namely between Timişoara and Klek. The new route was much straighter and favorable for navigation, but it depended a lot on the water level, whose flow constantly fluctuated between extremes.

Thus, the engineer Maximilian Fremaunt, in 1739, continued the sewerage of Bega, by intervening with levees to regularize the course of Bega. As a result, Timisoara benefited enormously from decreasing the flooding risk, but especially by drying of the surrounding swamps.

In the past, Timisoara benefited, through the Bega Canal (until the construction of the railway in 1957) from the only link with Central and Western Europe [30–33].

Between 1901 and 1916, the Bega waterway was equipped with modern installations (locks), which are still in operation, and in Timisoara the first hydropower plant in the country was put into operation. At the beginning of 1860, passenger transportation was going so well that in 1944, it peaked up to 500,000 passengers. The seat of the Port was the Iosefin district, on the section between the "Podul Muncii" (Labor Bridge) and the Gelu Bridge. Thus, the Bega Canal was connecting with Central and Western Europe, and heavy goods could be transported to Rotterdam.

The development of Bega continued at the beginning of the 20th century, when it experienced considerable development, by modernizing the locks, rebuilding the levees, and strengthening the banks. After 1944, the traffic on the Bega Canal began to suffer, and in 1945, the bridges in Timişoara and Otelec were removed from the riverbed, being destroyed by the bombings during World War II. After 1958, the traffic on the canal was impassable due to the war, where, this year, the transport of goods stopped, and in 1967, the passenger ships were withdrawn.

After the "Ceausist" Revolution of 1989 and the fall of communism, the reopening of the Bega Canal for navigation was also discussed, but the authorities at the time did not have funds for dredging the canal and resuming navigation.

In 2000, only the ship "Falcon" sailed on the Bega canal, which made recreational trips from the pier in the Children's Park to the Iron Bridge, the ship "Pelican", being used both as a terrace and as a pleasure boat, just like today.

With the support of the Dutch government, in 2002, the first feasibility study was elaborated, regarding the rehabilitation of the Bega canal, in order to make the canal navigable, a study aimed at reopening the traffic on the Romanian side. In the presence of the local authorities, on 1 October 2008, the works for unclogging and greening the Bega Canal began, a project to start from the Timisoara Hydroelectric Plant to the border with Serbia, over a length of 44 km.

Three or four years ago, Timișoara City Hall implemented a project on European funds, local funds and a contribution from the state budget, which had three objectives: arranging the banks of Bega mainly on European funds; creation of a second "bike sharing" subsystem of the city, containing 140 bicycles, on European funds; the purchase of seven vessels, on 100% local funds, and the introduction and maintenance of these seven vessels for regular public transport and recreation.

Therefore, since 2008, the ships have started to circulate successfully, to the delight of the citizens, on the Bega canal, recalling the former glory of the Port of Timişoara.

According to the Banat Timisoara Water Basin Administration, the Bega canal has a length of 44 km only on the Romanian territory, up to the border with Serbia and which once represented the pride of Timisoara. The length of the Bega Canal on the territory of the neighboring country is 72 km, often compared to the Seine River in Paris. Today, the city of Timisoara is known nationally, as the "city of Bega", being organized every year the Festival-Bega Boulevard.

## 3. Materials and Methods

The aim of this paper is to carry out a scientific research necessary for the development of integrated 3D maps by geospatial methods, to create a database by terrestrial 3D laser scans (fixed and mobile) for the Sânmihaiu Român Sluice located in Timiș County (Romania), with state-of-the-art technologies for creating 3D virtual realities. The aim of this work is to interpret and process LiDAR data, obtained using mobile mapping, with the help of specialized programs Pegasus Manager and Cyclone (Model option), in order to create a GIS database, which can be used by specialists in the field of hydrotechnics, designers and experts, whenever necessary. For this purpose, for the 3D scan of the Sânmihaiu Român Lock, situated on the Bega Channel, county of Timiș, Romania, scans were performed with both the ScanStation Leica C10 equipment and the Leica Pegasus Backpack equipment.

One may notice that this Leica Pegasus Backpack mobile scanning equipment is the first and the only one in Romania, the current research being pioneering in Romania.

On the Romanian territory, the navigation diversions are created by the Sânmihaiu Român Hydrotechnical Node and by the Sânmartinu Maghiar Uivar Hydrotechnical Node. U.H.E. Timișoara-Sânmihaiu Român Hydrotechnical Node, is a vital objective for the municipality of Timișoara, considering the functions of this work [34,35].

Currently, the Sânmihaiu Român hydrotechnical node ensures:

- maintaining a minimum level in the upstream buffer to ensure the supply of drinking, industrial, and fire water to the economic units in the municipality of Timisoara;
- achieving a constant level of diversion to maintain aquatic life and the ecological and health requirements of the population;
- the possibility of evacuating floods without flooding in the pond, as well as the evacuation of ice between 21 December and 21 March. At the same time, it ensures the maintenance of the debits transited within the limits regulated by the Romanian–Serbian agreement regarding the exploitation regime of the Bega canal, minimum and maximum;
- achieving the level for ensuring navigation on the canal pond;
- the possibility of evacuating, without flooding in the pond, the floods that form in the Bega river basin, on the UHE –Topolovăţ sector and the unloading of ice between 21 December and 21 March;

- the possibility of locking the ships (ensuring the transit between Sânmihai–Timişoara and Sânmartin–Sânmihai bays);
- the possibility of maintaining the flows on the Bega canal within the limits regulated by the Romanian–Serbian agreement on the operating regime of the Bega canal, respectively a minimum flow of 5 m$^3$/s and a maximum of 83.5 m$^3$/s.

Over time, there have been several actions to restore and secure the works. Thus, in addition to repairs and emergency interventions, in 1988, the capital repair investment of the hydrotechnical node was promoted. In a first stage, the necessary hydromechanical equipment and embedded parts were designed, manufactured and purchased.

The topographic surveys for collecting control points (GCP—ground control point) were made with the Leica GS08 GPS equipment, where for the realization of the project were also made topographic surveys, using classical measurement methods, using the total electronic station Leica TC802, where measurements were made to create and build a Bicycle Track, on a length of 36 km, starting from Iosefin Square to the border with Serbia.

The scan of the Sânmihaiu Român Lock, respectively of the entire Hydrotechnical Node with the same name was performed by Leica Pegasus Backpack equipment, where, by using SLAM technology (simultaneous localization and mapping) of ARTK (advance real time kinematic) technology to solve ambiguities and a high-precision inertial measurement unit (IMU) are required, making the Leica Pegasus Backpack a state-of-the-art "backpack", being provided inside with a 1 TB SSD computer, which ensures precise positioning, along with GNSS technology built into the equipment (ARTK—advance real time kinematic) and the Kalman filter. As a novelty, in addition to satellites, Glonass and Galileo can also receive signals from BeiDou satellites.

The number of satellites visible during the acquisition, at least five satellites are needed from the same constellation to count as a constraint over the trajectory (fixed position).

In order to perform the scans, one reference Master Station has been placed to collect RINNEX data at 1s, during the entire scanning session.

In order to complete the research and to obtain the "up-to-date" orthophoto plan regarding the Hydrotechnic Node of Sânmihaiu Român, respectively the Lock that is the subject of the work, were carried out for the comparison of data and air flights, with the help of a Drone Phantom 4 Pro (performed by the Chinese company DJI based in Shenzen), and the collection of 469 aerial images.

In order to georeference the UAV images, 12 GCPs (Ground Control points) were used (Figure 1), whose coordinates were determined using the Leica GS08 GNSS equipment. The processing of UAV images was performed with the AgiSoft PhotoScan Professional Edition program, version 1.2.2 (64 bit), where the GCPs were identified and marked in each image, manually, to achieve the most accurate georeferencing. Thus, it was found that the error of geroreferencing, using GCPs, respectively by checking them in the AutoCAD program, was between 2 and 4 cm.

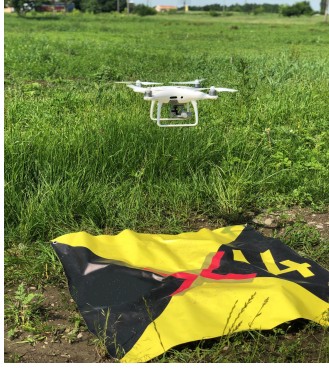 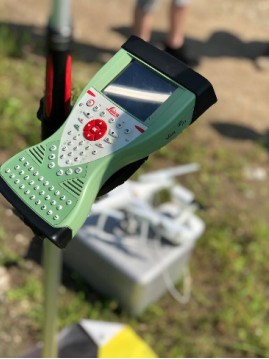

**Figure 1.** Presentation of UAV Phantom 4 Pro equipment, used ground control points (GCPs), of GNSS Leica GS08 equipment and the resulted orthophoto plan.

If in the case of engineering photogrammetry point cloud are obtained from aerial images collected with the drone (Phantom 4 Pro) and processed with the AgiSoft PhotoScan Professional program, in the case of mobile MMS scanning technology, where we used a Leica Pegasus backpack, point cloud are obtained from the two Velodyne type scanning systems [36], each one equipped with 16 LiDAR data acquisition lasers (LiDAR SO + LiDAR SA), at an accuracy of a few millimeters, and the images obtained from the 5 cameras will be used to color LiDAR dots, from the raw shape, which has bright red, yellow, and green colors, to a real shape, where each LiDAR dot will be colored based on the pixels from the images, in a split second. The same thing happened with the use of TLS technology, where a Leica ScanStation C10 equipment was used to scan the hydro technical objective, with the following clear differences (Table 1).

**Table 1.** Comparing scanning systems TLS–MMS.

| Features | Scanning Equipment | |
| --- | --- | --- |
| | **ScanStation Leica C10** | **Leica Pegasus** |
| Point cloud | 50,000 points/s | 600,000 points/s |
| Cameras | 1 camera | 5 cameras |
| Gyroscope | No, being static | Yes |
| Accelerometer | No, being static | Yes |
| GNSS | No | Yes |
| Possibility of choosing GRID | Yes (from 1 mm–X cm) | No (it is according to speed) |
| Data collecting time | Long time assigned/station, knowledge of placing and orienting the topographic instruments | Short time, being mobile, does not require targets and knowledge of placing in the station the topographic equipment |
| Processing time | Short enough, knowledge for automatic and manual alignment of the points | Long, SLAM processing, Kalman filter |
| Number of images | 260/station | According to the distance and the setup of the number of images/meters, which will be multiplied by 5 (5 cameras) |
| Device's preparation | Identic time with placing of the station in a total one | It involves assembling of a GNSS reference station, a dynamic initialization, and also, a static one, and after accomplishing scanning, repetition of the two initializations. |
| Use of targets | Yes | No |
| Ground control points (GCP) | No | Yes |
| Weight | 16 kg | 11.9 kg |
| Batteries' lasting | Big, with the possibility of change, during the scanning, it scans with 2 batteries at the time, and automatic permutation between them | Long, it scans with 4 batteries at the time, and automatic permutation between them, first the two upper ones will be consumed and then the two lower ones. |
| Scanning angle | 360 degrees in horizontal plan with 270 degrees in vertical plan | 360 degrees in horizontal plan with 200 degrees in vertical plan |
| Scanning temperature | Bigger than 4 °C or eventually, a, coat will be bought for the device, avoiding places with cold strong wind | Under 0 °C because the mechanism is covered, scanning being possible also in wind and cold conditions |
| Other optional elements | The necessity of producing Black and White targets for each assignment, or acquisition of 6 inch targets from providers | A GNSS equipment for RINNEX data collection, assembling a reference station. |

Both unmanned aerial vehicle (UAV) technology and terrestrial laser scanning (TLS) and mobile mapping systems (MMS) are important techniques for measurements and mapping. In the last years, UAV technology has gained tremendous interest both in the mapping community and in many other fields of application. By using digital cameras, UAVs is able to collect high-quality optical images for 3D modeling using photogrammetric techniques.

However, in the table below you can see the differences between the equipment and the comparison of some parameters made between DJI Phantom 4 Pro GNSS RTK [33], Leica ScanStation C10, and Leica Pegasus Backpack (Table 2).

<div align="center">**Table 2.** Comparisons between UAV, TLS, and MMS.</div>

| Features | DJI Phantom 4 Pro GNSS RTK-UAV | Leica ScanStation C10_TLS | Leica Pegasus Backpack_MMS |
|---|---|---|---|
| Control network | A lot of needs | A lot of needs | Only a few necessary in case when GNSS visibility is week |
| Regulations | Strict regulations, especially in urban areas | No | No |
| Interior/exterior | Only exterior | Interior and Exterior | Interior & Exterior |
| Data collection time | Decreased | Bigger | Decreased |
| Data processing time | Fast | Medium | Increased |
| Absolute precision in 3D | 5.0–8.0 cm | 0.1–1.0 cm | 2–5 cm |
| Flexibility | Medium (due to strict regulations) | Increased | High |
| Cost | Medium | Decreased | Increased |
| Price in EUR | ≈1500 | ≈80,000 | ≈400,000 |

Following the comparison of the obtained data, namely of the point cloud processed and obtained by acquiring the data with the three equipment, UAV, TLS, and MMS, the following aspects resulted:

- in case of using the UAV-Phantom 4Pro equipment, the orthophotoplan obtained, the orthophotoplan on the GCPs by checking them) for the Sânmihaiu Român hydrotechnical node;
- point cloud obtained with the help of Leica Pegasus equipment and mobile scanning technology, was between 2 and 3 cm for this hydrotechnical node;
- and the point cloud collected with the TLS-Leica C10 equipment, for the Romanian Sânmihaiu Hydrotechnical Node was between 0.1 and 0.7 cm.

In this sense, the point clouds obtained by the TLS were considered as a reference, where, for the three pieces of equipment, the resulting data were compared with all three methods having the coordinates in the Romanian Stereographic 1970 and Altimetric 1975 Black Sea projection system.

The georeferencing errors obtained after deleting the GCP that presented the largest errors and after performing the second optimization on the GCP, are presented in (Table 3). Ground markers removed were: GPS8, GPS10, and GPS15.

<div align="center">**Table 3.** Georeference errors of UAV images on GCP for NH Topolovățu Mic.</div>

| Photo | X Error | Y Error | Z Error | Error (m) | Error (pix) |
|---|---|---|---|---|---|
| GPS1 | −0.00082 | −0.05338 | −0.05578 | 0.086585 | 0.865 |
| GPS2 | 0 | 0 | 0 | 0 | 0 |
| GPS3 | 0.036718 | −0.01016 | −0.05006 | 0.066342 | 0 |
| GPS4 | 0 | 0 | 0 | 0 | 0 |
| GPS5 | 0.111514 | −0.10573 | −0.03657 | 0.166920 | 0.852 |
| GPS6 | −0.01769 | −0.05901 | −0.04148 | 0.051342 | 0.498 |
| GPS7 | −0.02703 | −0.05565 | 0.044331 | 0.089025 | 0.628 |
| GPS8 | 0.098082 | 0.206393 | 0.100105 | 0.294918 | 1.039 |
| GPS9 | 0 | 0 | 0 | 0 | 0 |
| GPS10 | 0.240903 | −0.05648 | −0.04699 | 0.284375 | 1.445 |
| GPS11 | −0.12949 | −0.1157 | −0.01983 | 0.143166 | 1.055 |
| GPS12 | −0.1046 | −0.04651 | −0.03665 | 0.101849 | 0.237 |
| GPS13 | −0.13328 | −0.11733 | −0.08312 | 0.169012 | 0.653 |
| GPS14 | −0.06569 | −0.00166 | 0.01723 | 0.057242 | 1.115 |
| GPS15 | 0.099543 | 0.055981 | 0.252045 | 0.529738 | 0.052 |
| GPS16 | −0.02441 | 0.050251 | −0.02056 | 0.048090 | 1.217 |
| GPS17 | 0.028593 | 0.170319 | 0.071616 | 0.197021 | 2.616 |
| GPS19 | −0.04183 | 0.071089 | 0.079123 | 0.12679 | 0.861 |
| Total errors | 0.041659 | 0.029480 | 0.042573 | 0.066461 | 0.607 |

The result, however, will not be the same as that obtained from a system equipped with Velodyne LiDAR sensors (Leica Pegasus Backpack) [36] (Figure 2) or a terrestrial laser scanning system (ScanStation Leica C10) (Figure 3), as they provide cloud point obtained from high-power lasers, which can deeply penetrate thus providing more detailed information about the surface area, as opposed to those obtained from aerial images.

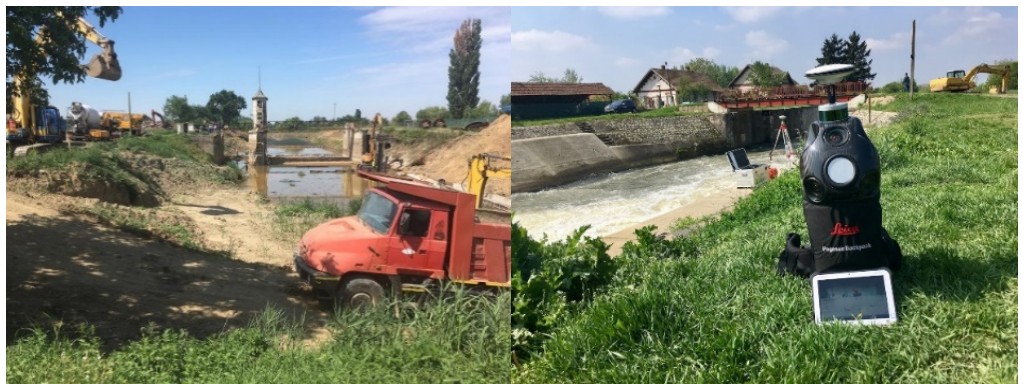

**Figure 2.** Scanning of the Sânmihaiu Romanian Lock under renovation using Leica Pegasus Backpack equipment.

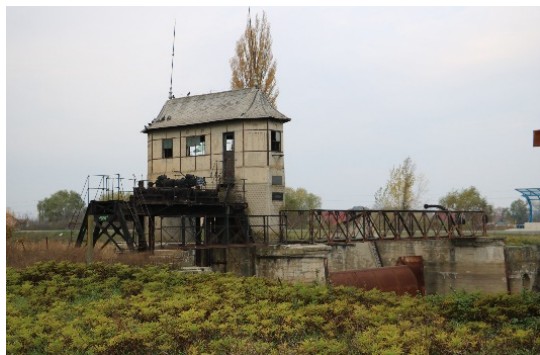 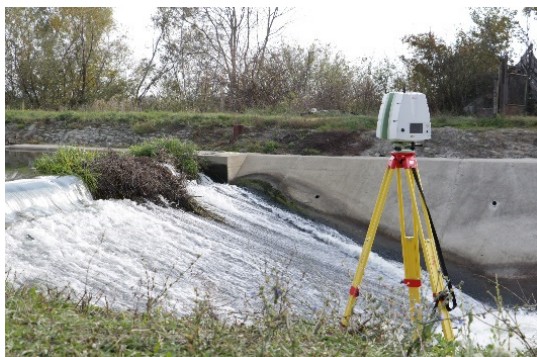

**Figure 3.** Scan of Sânmihaiu Romanian Lock under renovation using ScanStation Leica C10 equipment.

The combined use of this equipment leads to amazing results, complementing each other. Two additional 3D scans were also carried out using Leica Pegasus Backpack for the Hydrotechnical Node from Sânmihaiu Român, the restoration of which will be completed by 2021 (Figure 4), with a view to creating an up-to-date 3D database that can be used whenever necessary, including the creation of 3D heritage documentation.

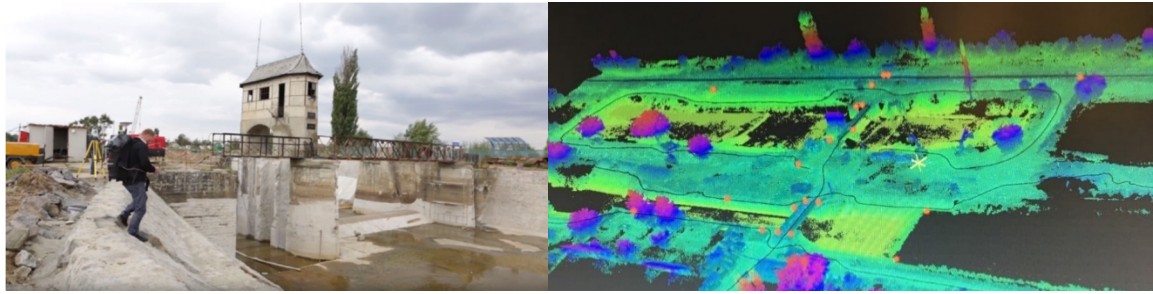

**Figure 4.** Scanning of the Sânmihaiu Romanian Hydrotechnical Node under renovation using Leica Pegasus Backpack equipment.

By integrating the camera with GNSS and IMU, it is now possible the automation of the process, in real time or post-processing, to "transfer" to the image a precise location, determined by GNSS. Previous photos are used in online map systems, such as Google Earth [37].

GNSS technology is [38], also integrated with LiDAR (light detection and measurement) technology, an optical remote objects detection technology, which at LiDAR frequencies is less than one millionth of a meter [39].

The Leica Pegasus Backpack is a portable system which combines two Velodyne VLP-16 linear scanners [40] (one vertical and one horizontal), plus five data texture cameras, a GNSS/IMU system integrated with Novatel SPAN technology (NovAtel's SPAN technology tightly couples our OEM precision GNSS receivers with robust IMU to provide reliable, continuously available, position, velocity, and attitude-even through short periods of time when satellite signals are blocked or unavailable) [41], batteries, and control unit (Leica Pegasus Backpack, 2017).

During data acquisition, a robust tablet shows the camera videos, profiles from the two linear scanners, and a diagnostic tool with information about the GNSS and IMU.

The equipment works with 4 batteries, two sets of two, with automatic switch between the sets, when the first set has less than 25% service life and with the possibility of changing a set of batteries during the measurements so that there is no need to carry out a new measurement project.

GNSS initialization requires a few minutes, called static initialization (of course in an open environment under the open sky, unobstructed by high vegetation or large constructions), followed by a dynamic initialization after which, it is necessary to take a few minutes' walk at a fast pace to calibrate the INS. When the acquisition path is entirely outdoors, a static GNSS purchase is required at the beginning and at the end; the dynamic session lasted throughout the period of scanning of hydro technical objectives.

Instead, when scanning takes place indoors, indications must start and close outdoors. If this start/close procedure is not possible, an appropriate platform can be used to re-locate the MMS at exactly the same point, i.e., to guarantee the correspondence between the starting and ending points, namely a "forced" closed loop.

A short stop ZUPT (zero velocity update algorithm) is required during scanning, being mandatory for indoor scans.

Aimed at overcoming cumulative errors and low positioning accuracy in single inertial navigation systems (INS), an optimally adjusted Kalman filter (OEKF) has been used in a closed environment. First of all, the errors of the inertial sensors are being analysed, modelled, and reconstructed. Secondly, cumulative errors in attitude and speed are corrected using the attitude fusion filtering algorithm, namely the ZUPT algorithm.

Post-processing data is done in two main stages: first, GNSS and IMU data are integrated to calculate the MMS trajectory (position and rotation of the SPAN platform) as a starting solution and later to calculate optimization, taking into account 3D profiles from both linear scanners. For the first step, inertial Explorer software (from Novatel) [42] is used, then the data is processed by Leica Pegasus software, namely Pegasus Manager.

The positioning solution is improved due to the contribution of images, through a similar performed by Zhang, J. and Singh, S. in 2015 [43].

In addition, the images from the camera are linked to 3D data for explorations and used for texture in the cloud. NovAtel's SPAN technology [15] combines precision GNSS receptors with robust IMU to provide reliable, position, speed, and attitude, continuously available, even in short periods of time, when satellite signals are blocked or unavailable.

In a GNSS system, the position is reported in relation to the phase center of the GNSS antenna. In an INS system, position, speed and attitude data are reported in relation to the IMU navigation center. An IMU is an electronic device that measures and reports a body's specific force, angular rate, and sometimes the orientation of the body, using a combination of accelerometers, gyroscopes, and sometimes magnetometers.

IMUs are typically used to maneuver aircraft (an attitude and heading reference system), including unmanned aerial vehicles (UAVs), among many others, and spacecraft, including satellites and landers). In order for a SPAN system to provide a combined GNSS + IMU position, speed, and attitude, it has to know where the GNSS antenna is placed in relation to the IMU.

IMU's forward direction orientation is also necessary to convert the perceived speed and attitude changes of IMU into the actual movement [44].

If the SPAN system includes other devices, such as a camera connected to an event entry, the SPAN system must know the location and orientation of these additional devices in relation to the GNSS. An INS is a system that is used to calculate the relative position over time from the rotation and acceleration information provided by an IMU.

An INS uses IMU measurements to ensure the calculation of position (INS = GNSS + IMU; GNSS fixes cumulative errors from IMU), speed and attitude (roll, step, and azimuth). When an IMU is combined with GNSS, the two navigation techniques increase and improve each other.

The stable relative position of the IMU can be used to go through the times when the GNSS solution is degraded or unavailable.

Leica Pegasus Backpack [45] is a reality-capture sensor platform with an extremely ergonomic design which combines five cameras that offer a fully calibrated 360-degree view and two LiDAR sensors (VLP-16) with a chassis of ultra-light weight carbon fiber.

It allows to obtain LiDAR points, both indoors (PURE SLAM method) and outdoors (FUSED SLAM method), at a level of precision that is authoritative and professional.

For scanning, in the case of research, the option FUSED SLAM was chosen, which allows scanning both outside and inside), compared with BASIC SLAM which assumes the option for scanning only indoors.

In the case of research carried out, the number of satellites visible during the acquisition, where at least five satellites are needed from the same constellation to count as a constraint over the trajectory (fixed position) [46], no compensation was applied using the SLAM algorithm.

This algorithm is used only on the inside, or if the lack of GNSS signal is weak and for a longer period of time.

For measurements starting from outside and containing part of the scan on the inside, having scan time less than 5 min, this option of clearing and adjusting the point clouds will not be used. Furthermore, outside, for "good" scans with more than 5 satellites and with a correct measurement trajectory planning, the error will usually include values between 2 and 5 cm as a scanning error.

For compensation, it will be taken into account, using the algorithm SLAM–the time.

How long was it scanned without a GNSS signal or less than 5 satellites? This time without a GNSS signal or with a weak signal will be subject to compensation, provided that this interval is bigger than 0.5–1 min [47,48].

## 4. Results

### 4.1. Acquisition of LiDAR Data Using MMS

For the acquisition of MMS data it was scanned Sânmihaiu Român Hydro technical Node using a FUSED SLAM method. This method is performed for both outdoor and indoor scans. This method uses a SLAM algorithm which combines GNSS/IMU navigation. The final trajectory and SLAM solution, under GNSS conditions, can be influenced by several factors, such as:

- Planning the mission of acquiring LiDAR data: using the Google Earth program.
- Equipment initialization: to obtain data with high accuracy, initialization was done next to the objectives which were scanned, in open sky area, not covered by obstacles (trees or buildings).

The initialization consists of two stages:

1.    STATIC initialization, for a period of 5 min, in which the backpack will stand still and can last up to 15–20 min in case that the ALMANAC data needs to update (Figure 5).

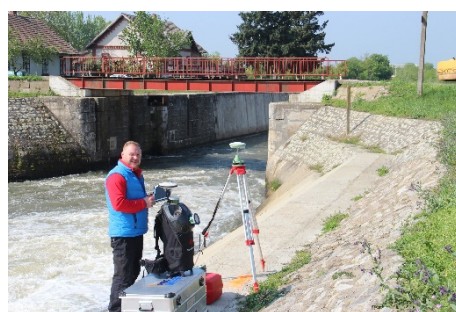 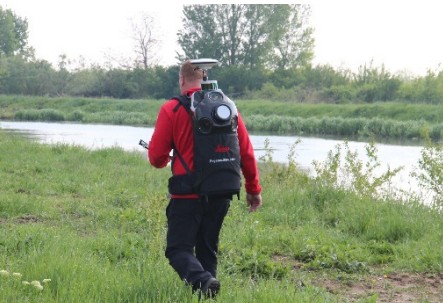

**Figure 5.** Static setup of the backpack for scanning and dynamic initialization-Sânmihaiu Român.

The GNSS almanac is a data set that every GNSS satellite transmits. It includes information about the state of the entire GNSS satellite constellation, as well as coarse data on every satellite in orbit. The orbital position of each satellite is known as the ephemeris data. While the ephemeris data helps deliver more precise locations to GPS devices, the almanac data helps the device locate a region of eligible satellites. This information is then saved for faster signal location in the future.

The GNSS almanac also includes clock calibration data as well as information for correcting distortions caused by changes in the ionosphere. When a GNSS receiver has current almanac data in memory, it can acquire satellite signals and quickly determine an initial position.

2.    DYNAMIC initialization, lasting 3–5 min, during which the backpack is carried on the back, performing a quadrangle, and at the end adopting the diagonals of the quadrangle until the message INS GOOD appears and the INS_GNSS will have a value below 1, usually the indicated value is 0.1–0.2 (Figure 5).

The ZUPT solution is an effective errors' adjustment way and it is calculated using an INS. In the algorithm, zero speed updating plays an important role, where zero speed ranges are detected and the speed error is reset. To use the zero speed update, it is necessary to reliably detect zero speed ranges.

Detection of the stop time interval is important for ZUPT methods. The ZUPT method, based on the Kalman filter, has a higher degree of precision compared to the ZUPT method, based on the mounting of curves and the ZUPT method based on estimating maximum probability, but Kalman filter modelling is essential for INS performance.

What is a zero speed update? Updating zero speed or ZUPT is a method of calculating the human step. If a person is walking, it can easily be said that there is a certain zero momentary speed when the foot comes into contact with the ground. Using this navigation information, based on IMU in a GNSS-free signal area is often done. IMU's costs suffer from noise, so navigation is not exactly accurate. There you can use Kalman filtering along with a ZUPT algorithm in order to correct the error.

A good ZUPT means parking for 20 s in an upright and still position. A good ZUPT will improve the SLAM solution and trajectory (Figure 6). Of the 20 s, the "Fine movement" option was chosen after processing MMS data. That is, a ZUPT almost motionless, and the Kalman Filter performed 55 calculation combinations, where out of the 20 s stationed motionless to achieve the ZUPT, perhaps two or three seconds of ZUPT were recorded perfectly motionless. Even when we breathe it is considered to be a moved ZUPT or if we talk to someone, for example, while performing the ZUPT.

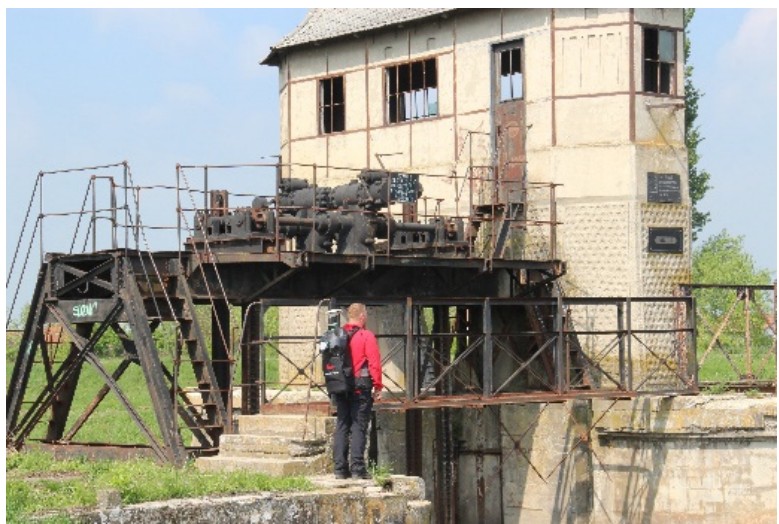

**Figure 6.** Performing a ZUPT outdoors during the scan (Sânmihaiu Român).

In outdoor and open-sky scans, it is not necessary to perform a ZUPT, but for the trajectory's improvement and for areas with high obstacles a ZUPT can be performed every 20 min of scanning.

Interestingly, during dynamic initialization a 2 s standstill at the time of dynamic initialization proved to be very useful both in the initialization and improving the trajectory, fact noticed in Sânmihaiu Român, when the first scan was carried out without 2 s of steady-state, and the second scan during the renovations taking place at the restoration of the Sânmihaiu Român Lock were used the two seconds.

Duration of measurements without GNSS signal: the time during which measurements were made without a GNSS signal will more or less affect the trajectory. For less than 5 min for the measurements taken outside there will be no problems, the processing will be simpler; ZUPT was also carried out on the inside, every 2 min, and also at each level climb, from one floor to another, the completion of the ZUPT will be carried out both before the climb of one level (floor) and when the other level has been reached;

The shape of the environment to be scanned: for areas where we do not have objects which might automatically build common points (a corridor longer than 15 m, without doors, windows or anything distinct) targets might be placed or chairs or anything else could be placed so that Mobile LiDAR technology recognizes the shapes;

The finesse with which the scan will be carried out: for this, during the scans of the heritage objectives listed above, the walks with Mobile Mapping System technology in the back were carried out at the same speed, without turning on the spot, without sudden movements on the left or right, which will influence the final trajectory;

Points' density: for this, the data is purchased at the same speed, without stops (apart from the moment a ZUPT is performed, if applicable), the change in speed will increase or decrease the number of purchased LiDAR points.

The workflow, using mobile scanning technology for the Leica backpack [49] "Mobile Mapping System" backpack is shown in Figure 7.

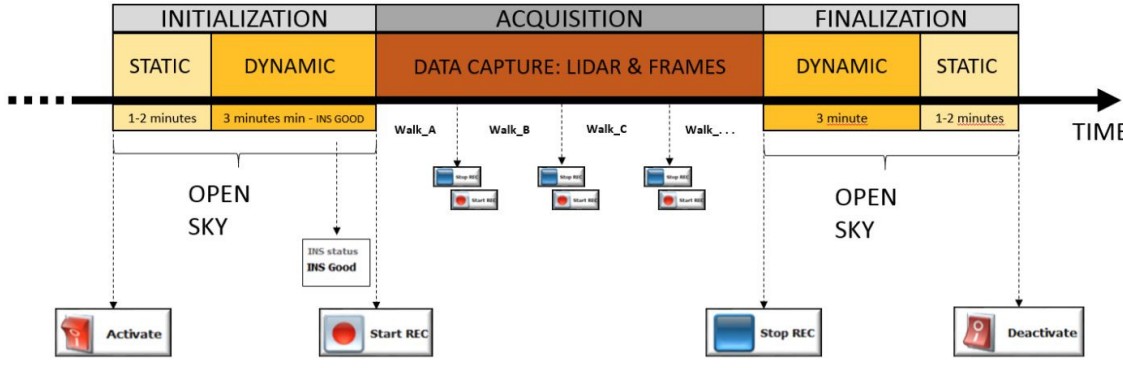

**Figure 7.** Workflow with Leica Pegasus Backpack (Leica Geosystems AG) [49]

*4.2. Acquisition of GNSS Data from Master Station and GCPs for MMS*

For the acquisition of LiDAR data for the objectives pursued in this paper, the FUSED SLAM method was used, method to which, in addition to the scanning backpack itself, we also need placing a Master station (a GNSS base) for field acquisition and RINEX data, which was set to an interval of 1 s to perform post-processing the measurement trajectory as opposed to static measurements for the storage of the geodesic network that was set to a purchase of RINEX data every 5 s.

It is mandatory to have GNSS data at 1Hz compared to the maximum location distance of the Master station which must not exceed a maximum of 15 km from the position of the Leica Pegasus backpack. GNSS reference stations located at the Cadastre and Real Estate Advertising Offices may also be used for a fee (3.5 EUR/hour/reference station). In carrying out the post processing for this work, an own GNSS Master Station type Leica GS 08 plus was used, which was mounted on a tripod for each of the measurements, in order to record RINEX data.

One mistake that should be avoided is to determine the height of the reference base (the Master station). The biggest problem encountered in the scans was at the end of the collection of RINEX data by the Master station. The same Master station point (which is usually a Feno-type landmark) was also read by the RTK process (in the Stereographic 1970 reference system, which is used in Romania), where, in the office, after importing RINEX data, the Master station is corrected with RTK data, and at the time of conversion, it seem that Stereographic data 1970 to WGS 1984 were not exactly accurate with the TransDat program, a coordinate transformation program carried out by ANCPI. A more precise transformation was carried out with the Leica Infinity program. However, the best solution was to record RTK data directly into the 1984 WGS system without converting it directly into the field or office.

For the correct collection of RINEX data, the Master station started 10 min before the backpack (Leica Pegasus Backpack) was initialized and was switched off 10 min after the LiDAR data collection was completed.

*4.3. Processing LiDAR Data with Leica Pegasus Manager. FUSED SLAM Method*

LiDAR data collection using the MMS Leica Pegasus Backpack [42] was carried out for NH Coșteiu, Sânmihaiu Român and Sânmartinu Maghiar Locks as well as for the Cruceni Pumping Station, using the FUSED SLAM scanning method.

4.3.1. Camera Calibration

The calibration of the cameras took place directly in the field, using the RADMIN program, installed on the tablet, where the color intensity balance was performed, in order to obtain clearer images.

4.3.2. Importing the Coordinates of the Master Reference Station and Processing the MMS Trajectory

In Pegasus Manager, for NH Sânmihaiu Român in the submenu Navigation Preparation takes place the import of RNEX data registration in the field with the equipment Leica GS08.

Navigation Preparation creates the project in the Inertial Explorer program and processes the trajectory using WayPointSDK.

The RINEX values of the Master station have been selected from the GNSS file, and the program will automatically load the latitude, longitude, and height on the ellipsoid expressed in meters. In the DATUM menu there were selected the type of coordinates: WGS 1984, where the database could only be set when entering Master station coordinates for PPP processing. "Preparing for navigation" converts the GNSS file from the imported Master station.

In the Advanced Options section, which contains three advanced types for processing GNSS and IMU data, i.e., Processing Times, Static Alignment, and Inertial Explorer (IE) option, the last option being chosen because it combines all possible computing possibilities.

In Option IE (Inertial Explorer) the useful settings for processing the trajectory will be done automatically, and in this case it is no longer necessary to define how GNSS Change IE Defalut data is filtered (high GNSS data collection quantity or high GNSS data quality) or fixation of ambiguities (accept all data, average compensation or high compensation, how to achieve ZUPT (slightly moved, fine or completely motionless) and the type of measurement compensation (pessimistic or optimistic).

If this option is selected IE Options, then manual trajectory processing is required and this will increase position accuracy as a result of the compass. Speed constraints have also been used in the trajectory calculation and apply to all 3 spatial directions. Automatic detection for ZUPT is also taking place: it is usually set to 0.5 deg/s. It is possible that, in case of problems, this value will be increased to 1 deg/s. However, in the development of indoor missions, ZUPT detection is a crucial step in determining the resulting quality of trajectory, since IMU has no information other than ZUPT.

For the interior the quality of a ZUPT is very important, because there was no GNSS signal during the scan, and the ZUPT is like a refresh of the scan.

*4.4. Post-Processing Data to Improve GNSS and IMU Quality in Inertial Explorer IE*

For applications that require a high level of precision and do not require real-time results, post-processing is a natural solution. Post-processing uses the ability to combine the results of the process back and forth, thus maximizing the accuracy of the trajectory and providing an indication of the reliability and accuracy of the solution.

Advanced processing techniques, such as smoothing back, may be applied in post-processing, which removes up to 95% of the position error against the full GNSS signal, compared to a real-time solution.

The visualization of the LiDAR scan trajectory processing, the accuracy of the data collection, and the measurement quality (QC Tools—Quality Control Tools) will be visualized in the Inertial Explorer program, using the graphs shown below.

A.   The graph for viewing the scan path using Inertial Explorer;
B.   Importing the trajectory into Google Earth [50], the processed trajectory may be viewed in Google Earth, and with the new function available on the epoch folder you can also see an altitude profile;
C.   The map for "Combined Separation", which shows us the quality of the scan, the quality of the equipment initialization at the beginning and end of the scan, the time during which the equipment did not have a good GNSS signal, or at all, and the error recorded in meters, also shows us the difference between direct and inverse solutions, in meters. You can also see the route in Google Earth [50]. It will also be determined here what time the SLAM compensation algorithm will be applied. Consequently, if in classical and modern topography compensations are made at angles and distances, this is the question of time, namely, "how long have we

measured without a good enough signal or without it entirely", meaning, "what is the time to be compensated", or in other words "how many seconds we have measured without a GNSS signal";

D.    Number of visible satellites at the time of scanning—this map shows us the time when, at the time of the scan, we had a smaller GNSS number or not at all, where the time will be taken as a value for post processing, namely the time interval without or with fewer satellites;

E.    The root mean square (RMS) that defines the continuous waveform.

These precision graphs will be presented below and have been made for the scanned objective: Sanmihaiu Român Lock (Figure 8), Sânmihaiu Român Site (Figure 9).

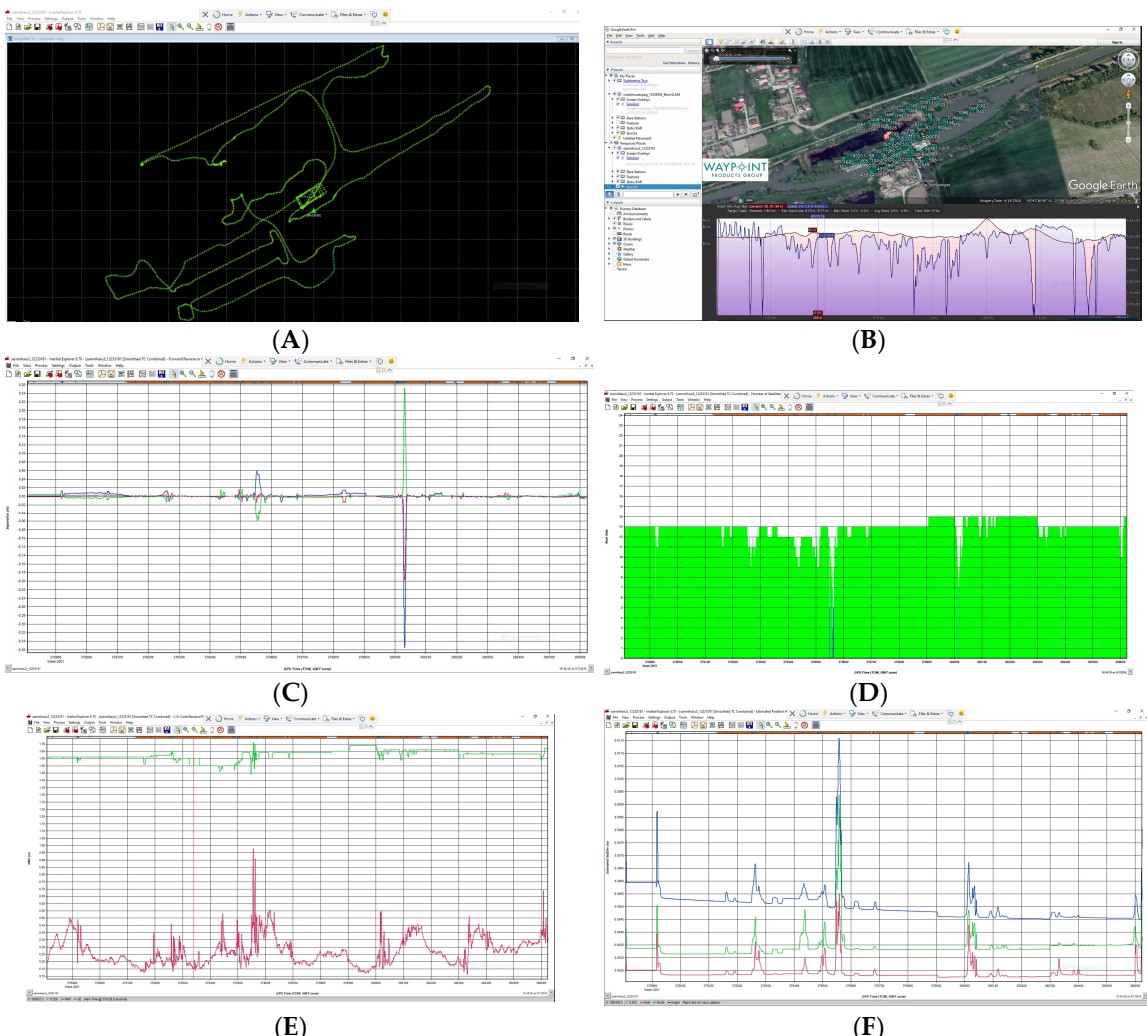

**Figure 8.** (**A**) Presentation of the scanning trajectory; (**B**) View the trajectory in Google Earth; (**C**) View the Correctness of the Scan (Combined Separation); (**D**) Number of visible satellites at the time of scanning; (**E**) RMS values; (**F**) Estimation of data accuracy (X, Y, Z and Time) for the Sânmihaiu Român Hydro technical Node on 2 May 2019.

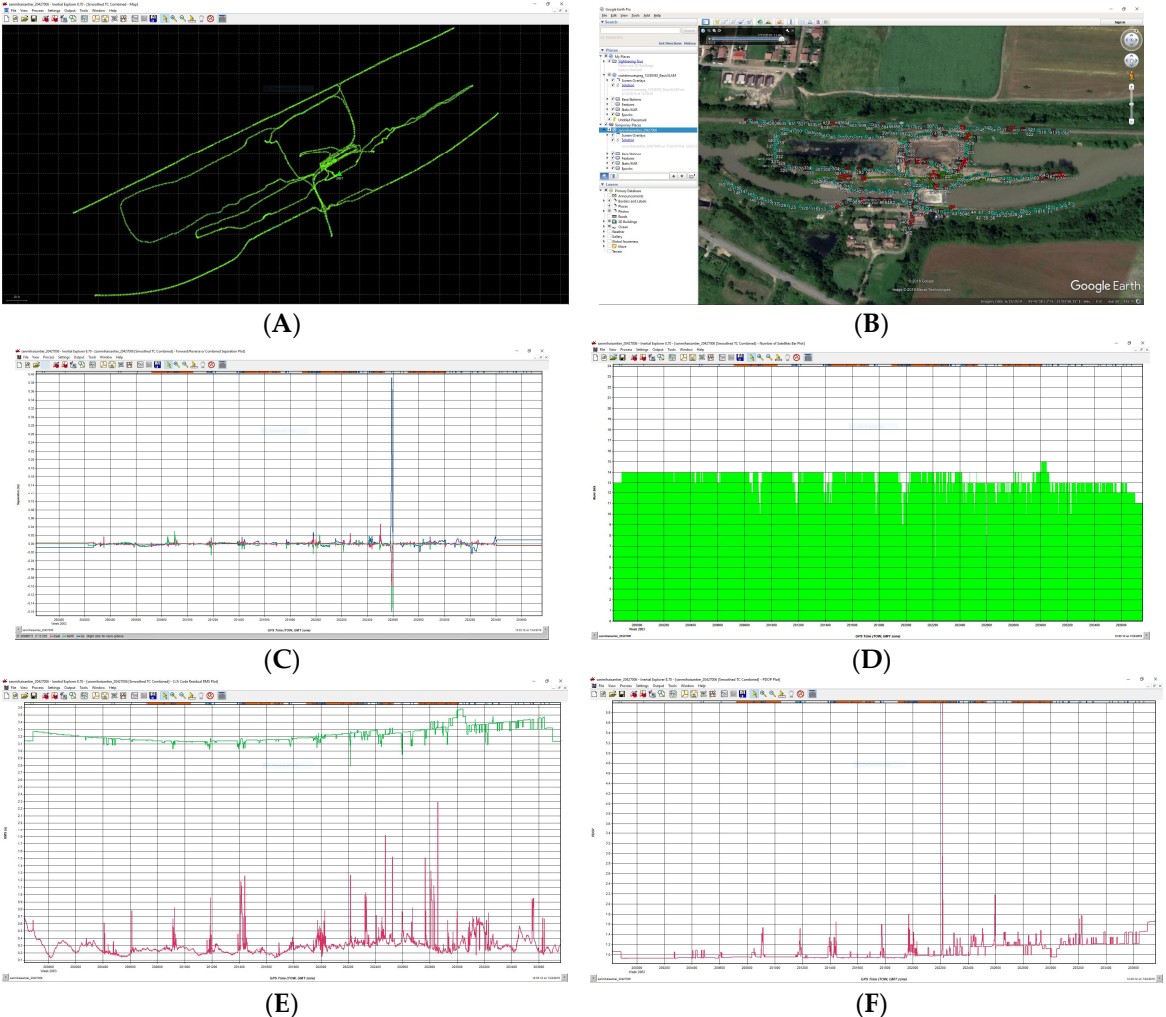

**Figure 9.** (**A**) Presentation of scanning trajectory; (**B**) View the trajectory in Google Earth; (**C**) View the Correctness of the Scan (Combined Separation); (**D**) Number of satellites visible at the time of scanning; (**E**) RMS values; (**F**) PDOP for the Sânmihaiu Român Hydro technical Node on 23 July 2019.

Within each group, the plots appear alphabetically in three colors: green, blue, and black. Green lots are generally the most frequently accessed plots, the blue parcels less, and the black parcels are rarely accessed, with the exception of advanced users.

Many plots support different units. For example, combined separation has been compiled, for the scan performed for the Sînmihaiu Român Lock, showing the difference between back and forth solutions in meters or feet. The separation of the distance or reference distance may be represented in units of kilometers, miles, or meters. To change the units in a graph, first it has been selected the graph from the list, and then the *Y*-axis tab was accessed. After the change of units, the preference was retained for all projects (Figure 9).

GrafNav/Inertial Explorer quality control graphs are organized into subgroups such as Accuracy, Measurement, and others.

These are some of the first data that will be checked to determine if the scan were performed correctly.

There is no map or graph showing us the true accuracy of time or location, but just some estimates to which a SLAM data processing algorithm can be applied.

However, for the most accurate scan, we will need to take into account the following: Number of satellites visible at the time of scanning: ≥5 satellites from the same constellation; Value for The

Combined Separation graph: <0.10 m and Fixed Ambiguity for Sânmihaiu Roman Lock (Figure 10);
Standard deviation: >0.05; PDOP: ≤3 (position dilution of precision).

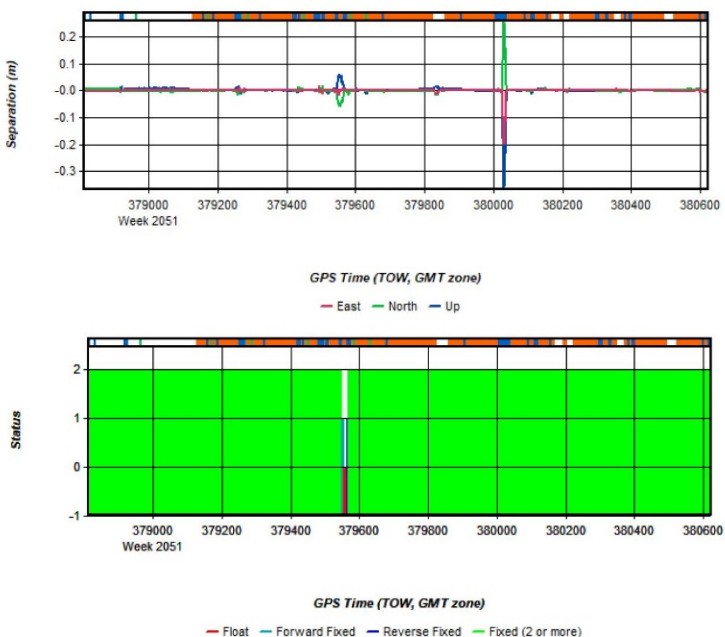

**Figure 10.** Combined separation and number of satellites for Sânmihaiu Român.

Individual parcels can be viewed by double-clicking a graph in the list or by selecting the OK button after selecting a graph. Up to two plots can be selected simultaneously, using the Ctrl key in combination with a mouse click before selecting the OK button. Furthermore, if a group of parcels was created using the Add Group button, all parcels within the group are represented simultaneously.

One may notice, from the image above for the Sânmihaiu Român Lock, that the scanning accuracy was between 1 and 5 cm, with one exception when the number of visible GNSS satellites was obstructed by tall trees for a period from 380,023.0 s to 380,036.0 s to a period of 16 s (Figure 11).

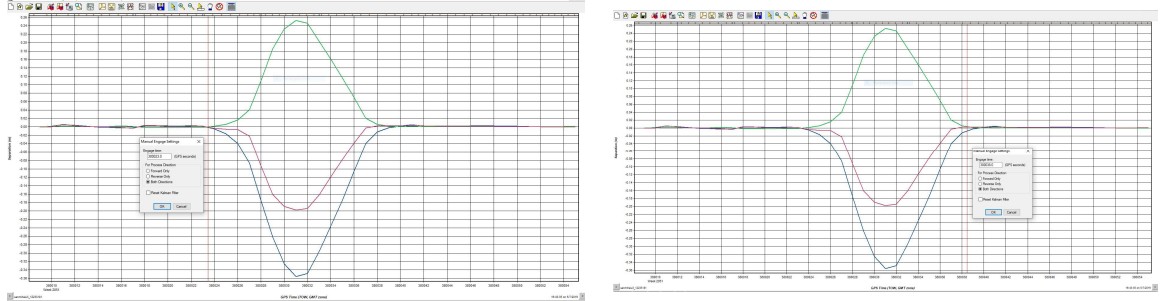

**Figure 11.** View the obstructed time of trees.

*4.5. Import Files from Pegasus*

When the scan files are imported, Pegasus Manager "unzips" the dot cloud from its original format to a readable format.

The workflow of this process can be easily changed, because a Pegasus system works with different scanning solutions, in this case the solution is "Pegasus system with Velodyne configuration" [36,40]. The end result of this process is the generation of an SDC file that contains the entire point of the part.

In this case, a Velodyne configuration [36] (Table 4) was chosen in order to clean the data. In this case, there is only the possibility to set "MIN INTENSITY (delete if it is lower)" which deletes the points with an intensity lower than the input value.

**Table 4.** Presentation of processing data for Velodyne SO and SA.

| Calibration | VelodyneBP54082_SO | VelodyneBP54082_SA |
|---|---|---|
| Angle Z | 55,000 | 65,000 |
| Angle X | −90,000 | −12,000 |
| Minimum intensity | 3 | 3 |
| Scanner model | VLP-16MX10 | VLP-16MX10 |
| SN Scanner: | 11001184574644 | 11001184426135 |
| FW scanner: | 3.0.38.0 | 3.0.38.0 |
| Scan Rate Index | 10.0 Hz | 10.0 Hz |

The value chosen is 3—"Create low resolution SDC"—which is a tool that generates a low resolution cloud.

This file is always required with the SLAM QC tool and the option has been checked by default. Creating low-resolution SDC files was omitted only for pure outdoor purchase. After completion, press "OK" to confirm and close. The minimum INTENSITY can be set to:

■   2 if the mission is very humid and there is a lot of water;
■   3 or 4 if there is morning fog or condensation;
■   5 to 8 for normal and sunny days; this value can range from a minimum intensity of 3 to 50.

### 4.6. Extracting JPG Images

There were 1725 images extracted from each camera (1725 × 5 cameras). Each camera was set to take pictures every meter. If we do not want the images to be so dense, and the raw file not to be exaggeratedly large, we can set the five cameras to take pictures every 3 m or every 5 m.

### 4.7. Time Alignment in Pegasus Manager

Usually, the software [9] automatically matches the GPS time and the trigger time of the PC. By default, if the software cannot automatically match both times, a window appears, where you are asked to manually adjust the PC trigger time and GPS time.

Fix time alignment—If this is not enough to align correctly, the user must add the following section to the GVS.ini file:

[Time Align]; default = 0.06 = 60 ms DeltaTime Match = 0.06

In this case, the solution offered was Ok, and in the end it was no longer necessary to match the delta time, gradually increasing this value (which is default 0.06). If the default value does not provide a good time alignment, then the value will necessarily be increased, using 0.065 then 0.07, 0.08, etc., until a good percentage is obtained for time alignment (almost 100%).

### 4.8. Import Trajectory (Import Trajectory Data)

In this part of the program the trajectory was loaded (for the Romanian Sânmihaiu Lock, but also for the other scans carried out in the framework of the research) automatically, a trajectory resulting from the preparation for navigation *(Navigation Preparation)*. This is also where the new, manually processed trajectories will be loaded. The trajectory from basic and advanced SLAM offsets can be automatically charged.

The E-TransDatRO coordinate system with the following data will be loaded in the coordinate system submenu:

Transformation: None

Ellipsoid: Krassowski

Projection: Romania Stereo 70; Geoid Model: EGG97&MN75

CSCS Model: csRomania; Source: E-TransDatRO.csys (Walk_A- Import trajectory data

Converting IE file . . .

Trajectory

D:\Pegasus Work\Prelucrari\SANMIHAIU ROMAN-2\2019-MAY-02_sanmihaiu3\GPSINS\sanmihaiu3_12233181.cts

Interval: 0.008

Inertial Explorer v8.70.6404

GPSINS folder: D:\Pegasus Work\Prelucrari\SANMIHAIU ROMAN-2\2019-MAY-02_sanmihaiu3\GPSINS

Rover: 12233181

Loading tracks information . . .

Calculating quality factor . . .

LeicaPegasusNavPrep: Done!

Writing File . . . Done!

IE LOG file= D:\Pegasus Work\Prelucrari\SANMIHAIU ROMAN-2\2019-MAY-02_sanmihaiu3\IETRJConvert.ie

YAW reference = 396.895236 grad

IE to TRJ—done

Min quality value: 0.930

Max quality value (95%): 1.300

Max quality value (100%): 1.490)

### 4.9. Image Orientation (Generate Image Orientation)

At this stage, the AVP (ArcView Data) file is generated, the errors are automatically fixed, the orientation of the 5 cameras is exported and the KML (Keyhole Markup Language) file for the 5 cameras is created (Figure 12).

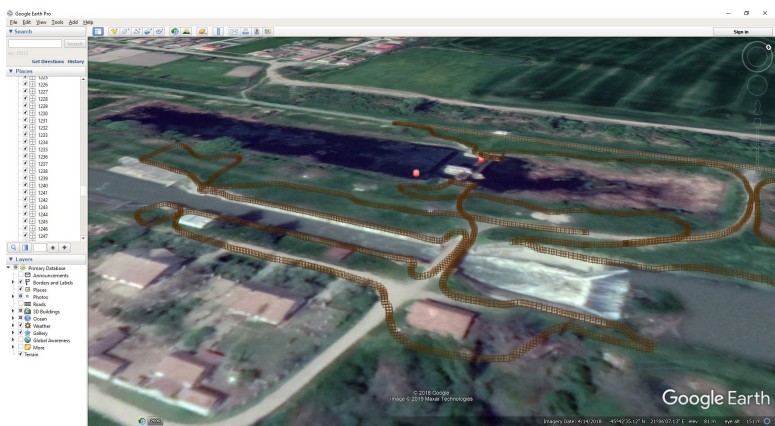

**Figure 12.** Image orientation and export of KML file for Sânmihaiu Roman Lock.

### 4.10. Generating 3D Point Clouds

The following data will be taken into account at the 3D point cloud generation stage:

- The maximum distance of processing of Max point clouds that will help us to process points on a set width, for example for the Sânmihaiu Romanian Hydrotechnical Node, a width of 40 m and 1.35 m min was selected.
- Maximum number of points—set to 20 M (20 million points).
- Voxel size was set to 0.0005 m—that is, a point has a size of half a millimeter.
- Point cloud coloring—which has been selected so that you can see RGB (Red, Green and Blue light) point clouds.

And last but not least, the masking of the GNSS antenna in the images and their processing will be taken into account (Figure 13). Once the antenna is masked from a single image (Figure 14), it will be automatically masked from all images. The number of images that will be collected at the time of scanning can be set from 1 picture/m to 5 pictures/m. For Sânmihaiu Român, one picture/m was selected for each room, and for the Cruceni Hydrotechnical Node, one picture for every 3 m was selected.

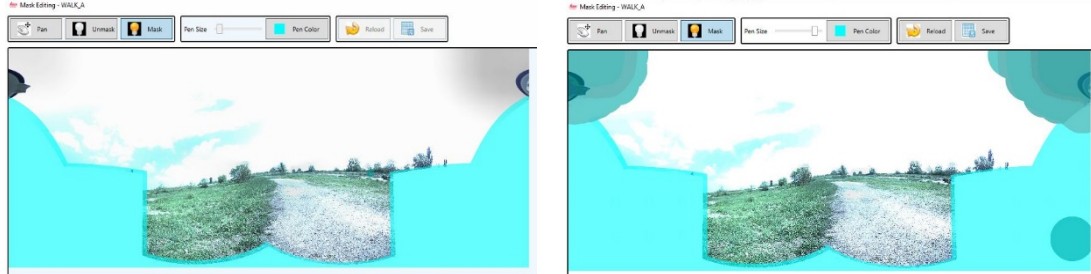

**Figure 13.** Before and after masking the global navigation satellite systems (GNSS) antenna.

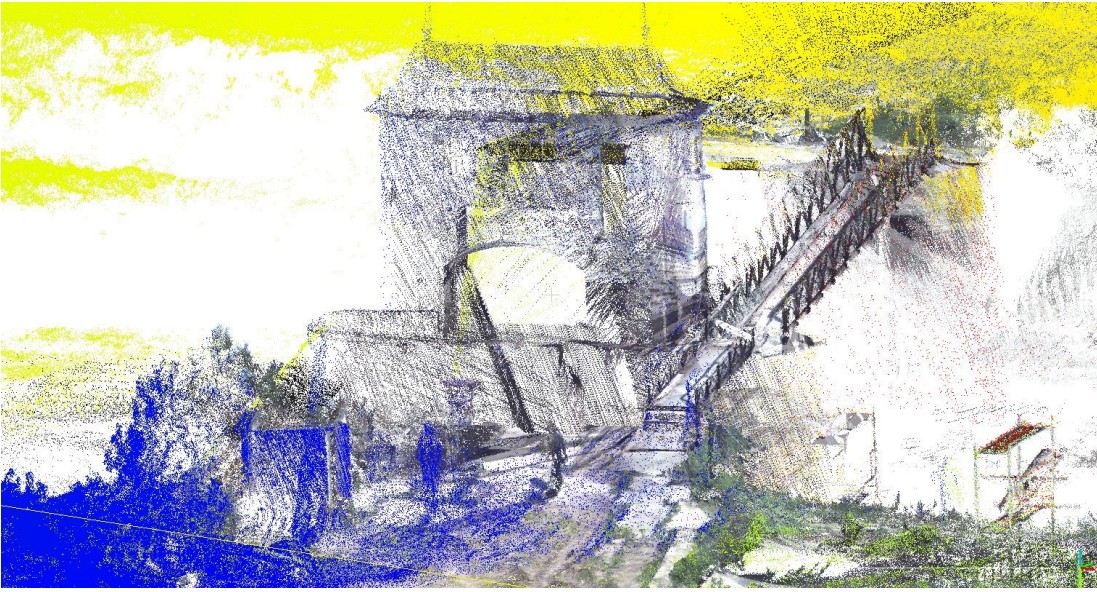

**Figure 14.** LiDAR data obtained by Velodyne VLP16 scanning at Sânmihaiu Român.

This is followed by data processing and obtaining 3D point clouds.

### 4.11. Creating Peg Tracks (Foot Track)

When scanning the heritage hydrotechnical objectives, in order to remove from the scan the footprints left by the legs while walking with the backpack in the back, the height of the IMU was set from the height of the road to 1.35 m, and the minimum distance between the points of 0.1 m.

### 4.12. Point Cloud Filtering

Filtering type, which can be: none, decimation, smoothing, decimation + smoothing, smoothing + decimation and decimation + smoothing + decimation. For the present paper, the NONE type was chosen, i.e., none; Noise reduction, which was set at a cell size of 0.05 m; The minimum number of points per Voxel was taken as the value 2.

Filtering results were not as desired, where from the point of view of point clouds, filtering is not performed correctly for this type of Velodyne LiDAR VLP16 scanner [36,40], because LiDAR data, or better said point clouds are not placed in a straight line, as in the Leica C10 but are arranged in an X (Figure 14). Leica Pegasus Two, the one fixed on the machine achieves a very good and successful classification of point clouds, and data processing in 3D mode is much simpler.

### 4.13. Voxelization of LiDAR Data

Moreover, in LiDAR data there are usually exceptional values, resulting from the multiple reflections of object structures, such as trees, the uneven reflection characteristics of objects themselves, such as buildings, and reflections of birds or suspended objects at higher altitudes.

Accuracy and efficiency are largely influenced by excessive costs. A histogram examination technique shall be used to avoid the remote effect. A histogram shows that the general height distribution characteristics are generated. The height points are determined by visual evaluation to remove the smallest and highest tails. LiDAR points that are higher or lower than the highest ($T_h$) or lowest ($T_l$) elevation thresholds are removed from the data set [51,52].

### 4.14. Generating Spherical Images and Exporting Files

It is an optional operation within the program "Pegasus Manager", but for the realization of this research was realized the generation of spherical images and thus obtaining stereographic images.

The 1725 images will be taken into account for the processing of spherical images.

Export of point clouds has been performed with the LAS extension, but TopoDOT 10.5.0.17 extensions can also be exported; Recap 3.0.0.52, E57 1.3, Depth images and PTX, Pegasus Web Viewer and you can blur people or machine numbers with The Blur make and car license *flat,* which is also used by Google.

### 4.15. Introduction of GCPs

The introduction of checkpoints for additional scan verification and data processing will be viewed in the "View Process Data".

Importing of GCP into the Pegasus Manager program for Sânmihaiu Român Hydro technical Node is shown in Figure 15.

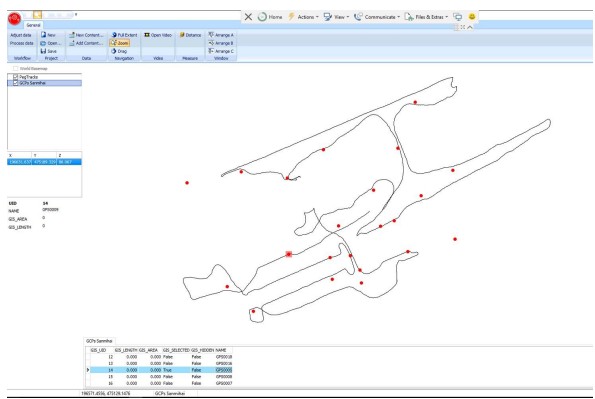

**Figure 15.** View GCP checkpoints.

### 4.16. Video Catalog

A video catalog contains the links to the trajectory databases that were included, in the case of this research, in Leica Pegasus Manager for Sânmihaiu Român Sluice. All images and video files for the selected tracks have been uploaded directly, and the profile files will be read only after the LiDAR dataset is defined. There are two ways to define a video catalog:

■ automatic—this is usually configured through the Configuration Wizard. The Video Catalog file is called Videocat.dbt and is stored in the first project folder used in processing;

■ manual—this option is used to specify the existing video catalog to be uploaded to Leica Pegasus Manager. It could even be used to create and save a new catalog that includes one or more paths.

When creating a new video catalog, all the data that will make up this catalog must be organized. The path files must be on the same hard disk.

In the Video Catalog tab of the "PEGASUS Manager-Configuration" window, two parameters define the required cursor distance in the path to open the Video window:

■ The piece search radius is how far the software [53] will search for a point on the path, starting from the cursor position in the Map window.

■ DMax, is the size of the side of the cube used to display Leica Pegasus data around the cursor in the Map window.

"Video" on the "General" tab contains the "Open Video" button, which can be activated even on the same icon in navigation tools in the upper-left corner of the window (Figure 16). You can also measure a distance here using the "Measure" box and contain the "Distance" tool [54].

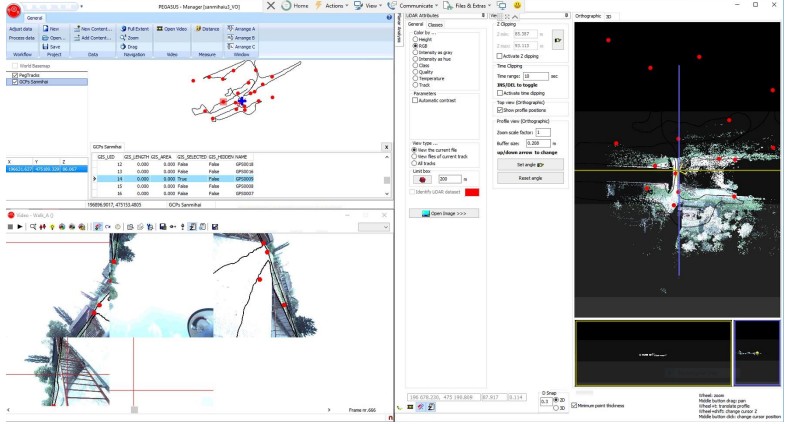

**Figure 16.** Automatically open of the video catalog for Sânmihaiu Român.

Three types of window arrangement choice is available. Type C arrangement shown in the top figure, type A and type B.

■ Type A Arrangement—The window is divided horizontally into two parts, the upper half is divided in two, on the left is the Map window, on the right the LiDAR window, and at the bottom you can see the Video–Picture window.

■ Type B Arrangement—the window is divided vertically into two parts, the left half hosts the Map window, while the right half is split in two, at the top is the LiDAR window, and at the bottom is the Video window image.

■ Type C Arrangement—The window is divided vertically into two parts, the left half is divided in two, at the top is the Map window, at the bottom you can see the Video–Image window, while the right half hosts the LiDAR window.

### 4.17. Viewing the Obtained Point Clouds

By simultaneously scanning the same object, the alignment of point clouds takes place, which, as presented above, may be correct, or may be wrong, at which point the SLAM compensation and alignment algorithm will be used. This will automatically create landmarks from the first position, and in the second scan will identify the same common points, and if there is a trajectory error it will automatically adjust to the common points of the two scans and realign the trajectory and point clouds.

Figures 17 and 18 represent the point clouds for the 3D representation of the Sânmihaiu Român Lock, scanned on 2 May 2019. The Sânmihaiu Romanian lock was scanned once more, during the repairs on the 23 July 2019, and the 3D model is represented in Figures 19–21 where you can see an overall scan of the Sânmihaiu Român construction site.

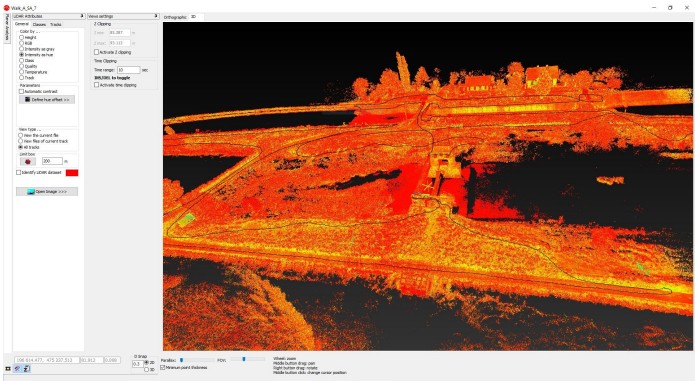

**Figure 17.** Intensity of LiDAR data nuance, Sânmihaiu Roman Lock, scan performed on 2 May 2019.

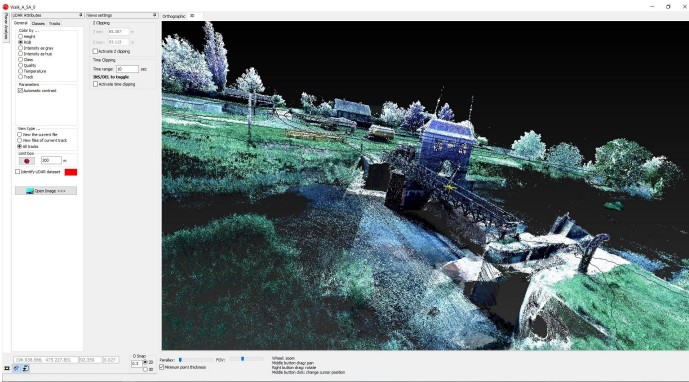

**Figure 18.** Viewing LiDAR point clouds obtained by scanning the Sânmihaiu Roman Lock (RGB).

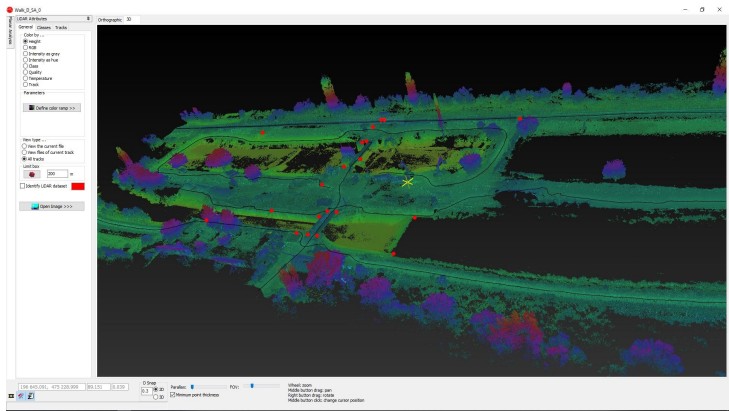

**Figure 19.** LiDAR data presentation, colored by altitude, NH Sânmihaiu Roman, construction site.

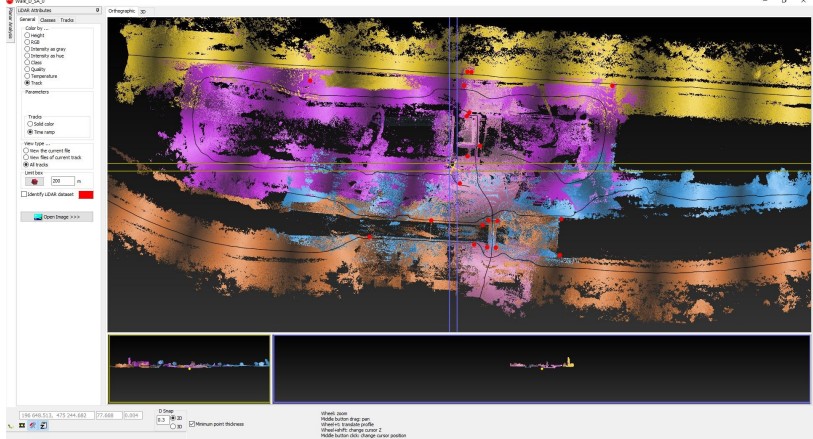

**Figure 20.** Coloring of the points clouds by the Walks, NH Sânmihaiu Roman.

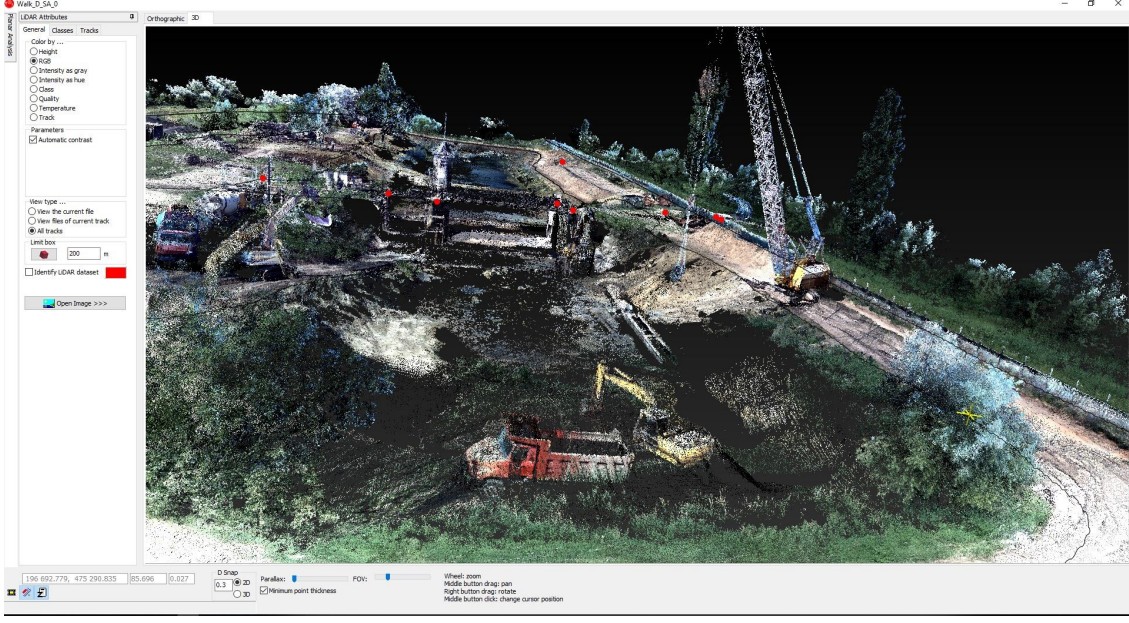

**Figure 21.** Restoration of Sânmihaiu Roman Lock, performed on 23 July 2019.

The intensity of the color shade, the LiDAR point clouds colored by height, and the visualization of the RINNEX Master data collection station for Sânmihaiu Roman will be presented in Figure 22.

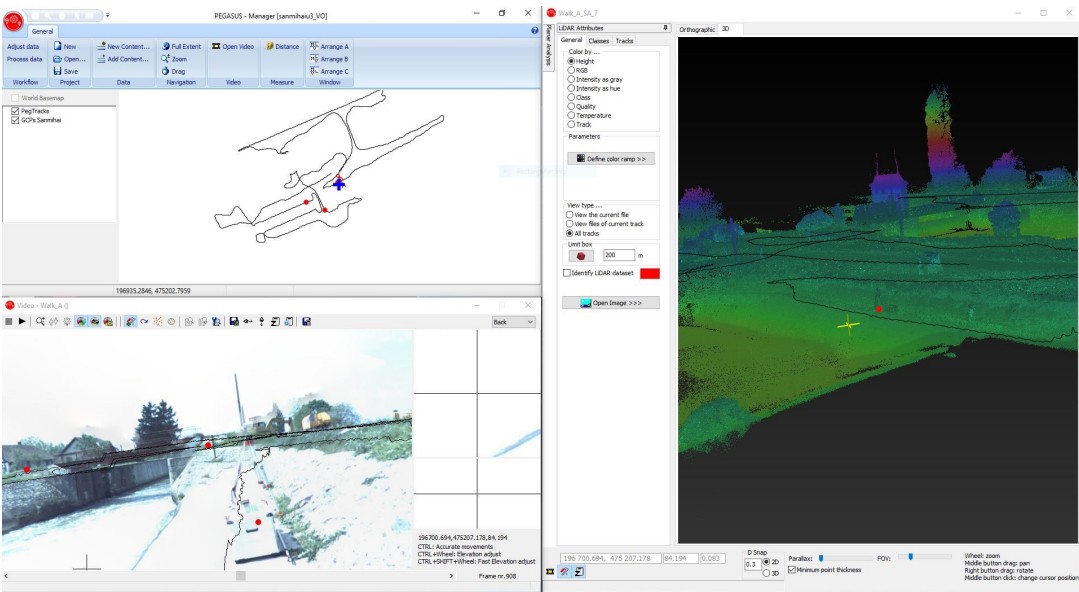

**Figure 22.** LiDAR point clouds and their height coloring.

Spelling view involves viewing the database in a classic three-window view, in the top half of the plane, and in the lower half; on the left is the cross section and on the right the longitudinal section.

3D viewing involves free navigation in the 3D cloud database.

### 4.18. Using Processed Video Images for GCP

The video image window was used to display images purchased from one or more cameras, from locations scanned for research (Figure 23), as well as to upload the associated profiles of the laser scanner and even navigation, using the "Video Image" window, to navigate through the measured path.

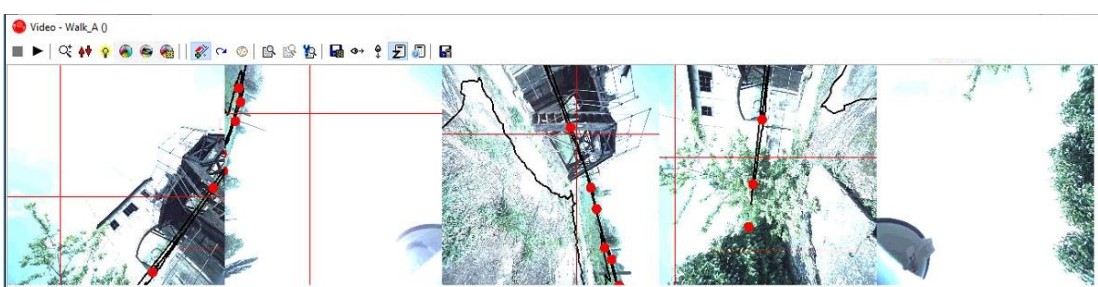

**Figure 23.** Video images of Sânmihaiu Romanian Lock.

### 4.19. Use of Spherical (Panoramic) Images for GCP

The "Spherical Images" button opens a new window showing spherical images. The window is arranged by a main window on the left and two pan windows on the right, one above the other (Figure 24).

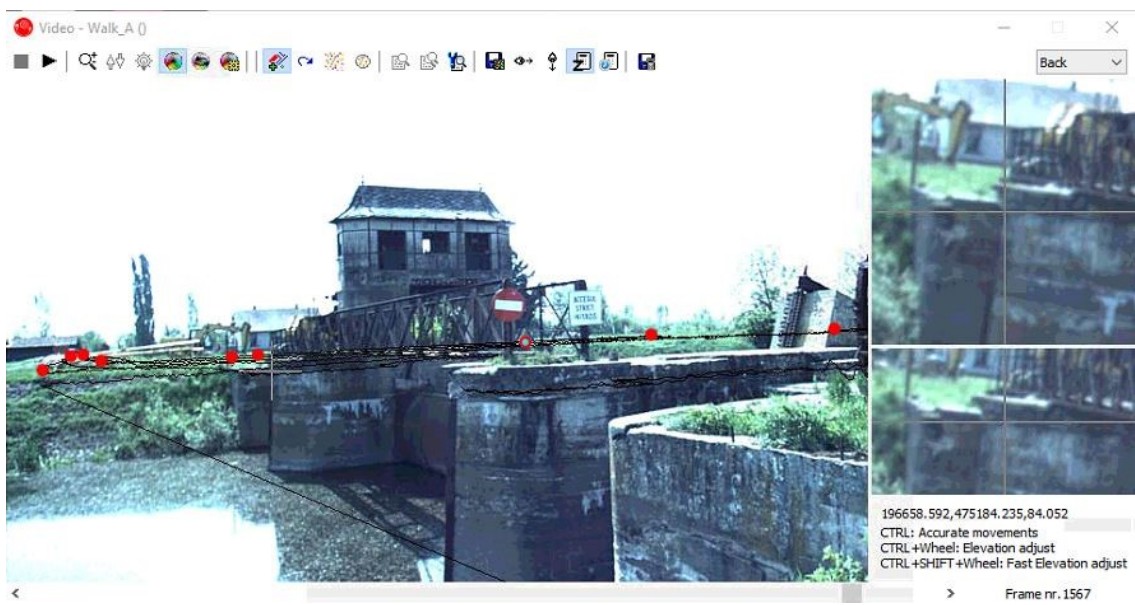

**Figure 24.** Spherical images of Sânmihaiu Romanian Lock.

### 4.20. Viewing LiDAR Data on Stereographic Images

In the Video image window, the spherical images for the Romanian Sânmihaiu Lock were opened, where a camera was selected, after which the window containing the point clouds (Point Cloud) was opened.

The 3D representation of LiDAR points superimposed over the vacuum image is shown in Figure 25 and contains all settings for viewing the point clouds of images.

- ■ "Point size": defines the size of each point of the point cloud. A lower value makes the image behind the dots more visible.
- ■ "Density": is the percentage of points in the cloud that will be moved on the image. A lower value reflects a smaller number of points that will be viewed in the "Video Image" window.
- ■ "Transparency of Points": A higher value makes the cloud transparent and the image behind the cloud more visible.

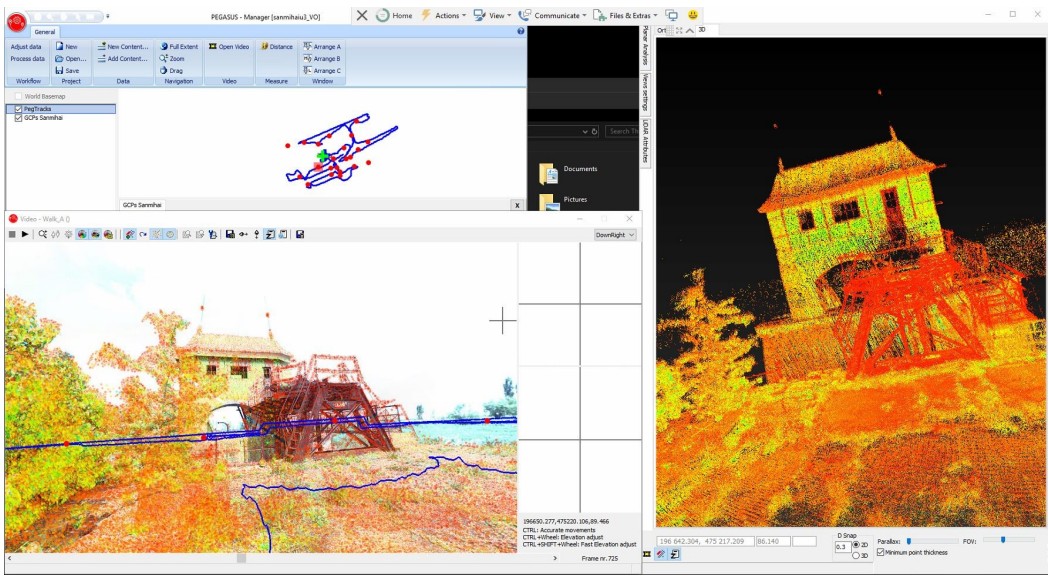

**Figure 25.** Presentation of LiDAR points over the video image and 3D presentation of LiDAR points.

### 4.21. View LiDAR Points in Real Time

This tool is particularly useful to be able to do a detailed analysis of the overlap between the point clouds and stereographic images.

The figure below (Figure 26) defined the point size in the LiDAR point clouds.

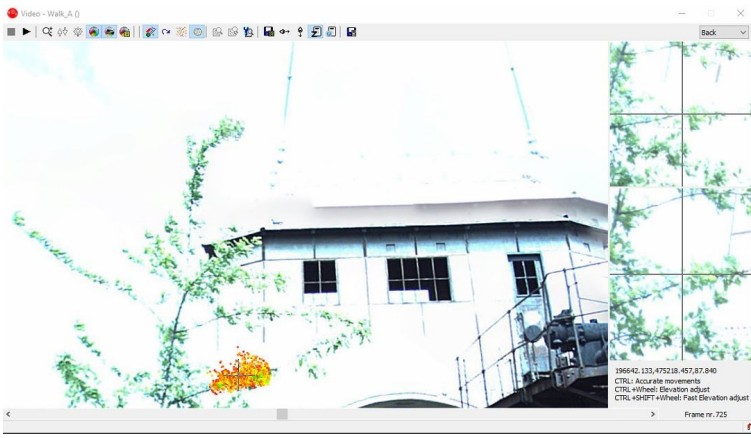

**Figure 26.** Real-time view of point clouds.

### 4.22. 'Parallax' Slide

The "Parallax" slide was used to change the parallax for 3D viewing, depending on your preferences. To view the point cloud in 3D mode, using a stereoscopic monitor, you will be able to use the "Anaglyph" mode or you will be able to use tools that allow a stereoscopic vision (Figure 27).

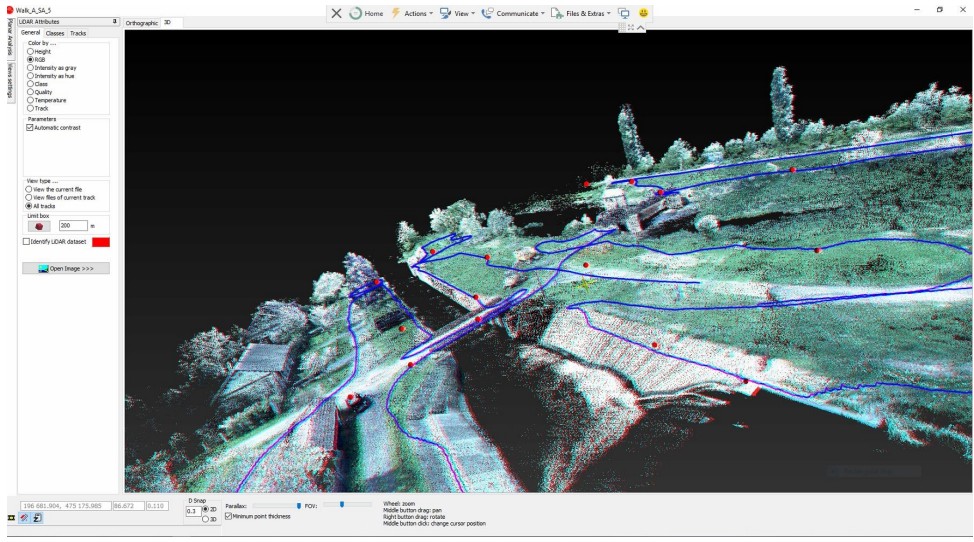

**Figure 27.** Parallax for viewing using a stereoscopic monitor.

### 4.23. Coloring of Point Clouds after GCP Verification

The color is used to define the ways of viewing the data set on the screen, depending on the different filters and classifications available. Color variation can be really useful to easily highlight the information you want on the screen. It can be colored in different type:

- colored based on the Z coordinate of the points (Figure 28);
- colored considering the RGB values (Figure 29);
- colored according to the gray intensity of LiDAR point clouds;

- colored according to HUE intensity;
- colored according to the 3D quality of the points;
- colored by "walks".

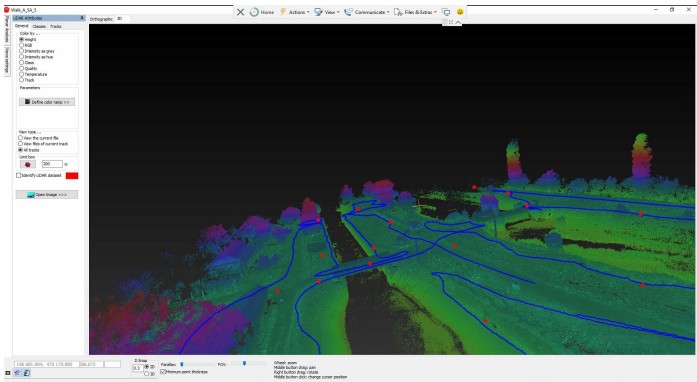

**Figure 28.** Coloring of LiDAR point clouds by height.

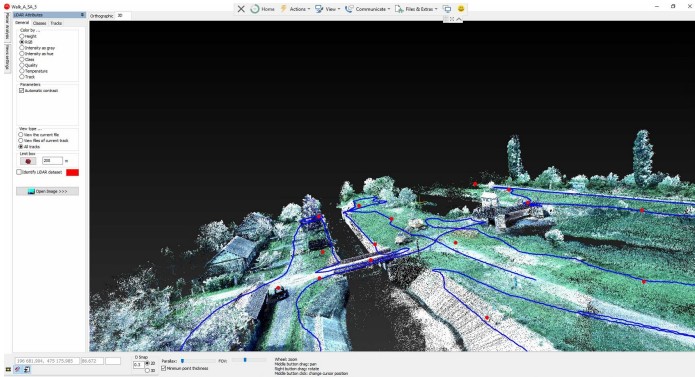

**Figure 29.** RGB coloring of LiDAR point clouds.

*4.24. Trajectory Adjustment on GCP Checkpoints According to Time*

The GCPs will be imported and set (Figure 30).

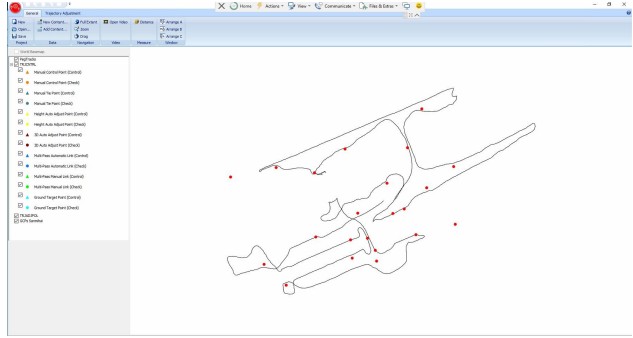

**Figure 30.** Import and visualization of GCPs, Sânmihaiu Român Lock.

In the video sub-menu, TRJCNTRL (Trajectory Control) was selected on the left, at Layer, and on the right, in the Layer sub-menu, GCPs were selected (Figure 31). Type C viewing has been arranged for better viewing.

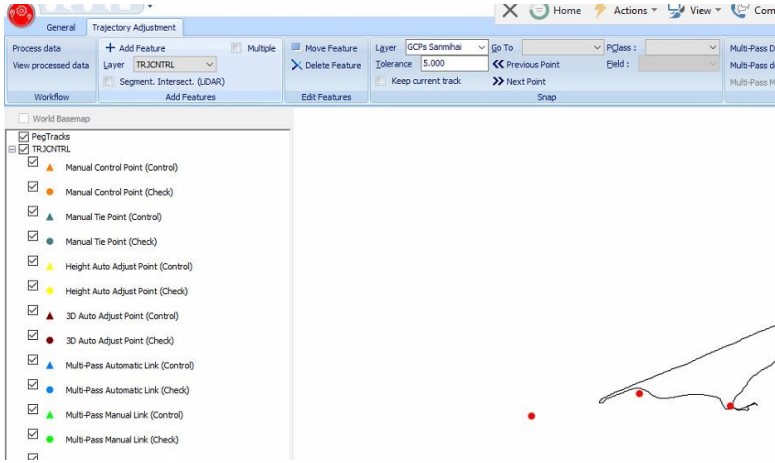

**Figure 31.** LiDAR measurement trajectory control.

Next, the video option will be open in order to view the GCP checkpoints and adjust them according to time (Figures 32 and 33).

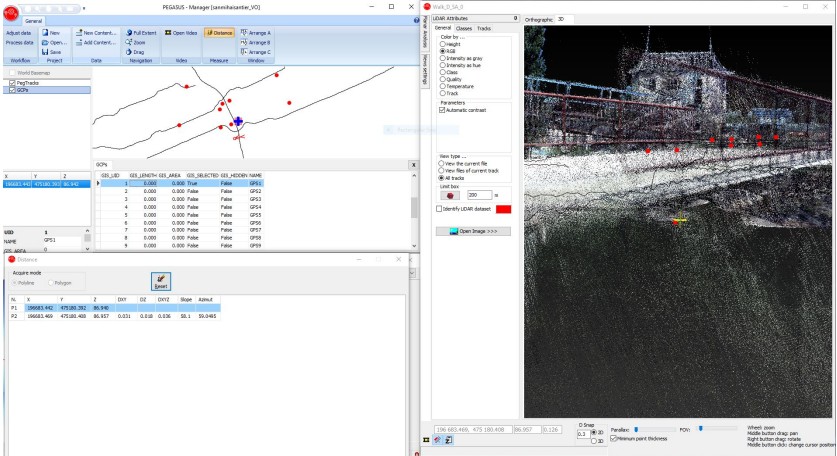

**Figure 32.** Adjustment on GCP No.1.

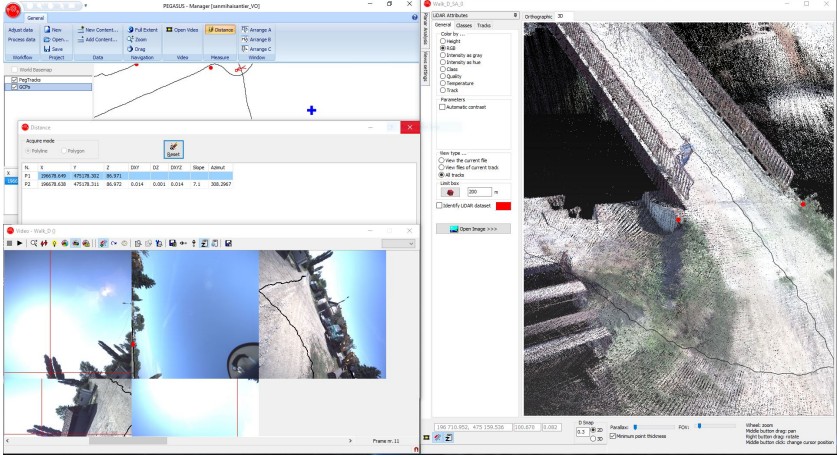

**Figure 33.** Adjustment on GCP No. 2.

A very important step in terms of the quality of collected data using MMS technology, was also the field collection of GCPs, which were also determined with the Leica GS08 plus equipment, using the RTK method.

Verification of the checkpoints (CP) was performed by creating a database and importing the coordinates of the checkpoints into the Pegasus Manager program.

After importing the GCPs into the Cyclone program, position analyzes and 3D measurements were made to compare the data (LiDAR point cloud compared to the GCPs).

If the GCPs do not match on the scan), time or position compensations will be made for the LiDAR point cloud. All control points were checked during the work. Further research will present the coordinates of the GCPs studied, the quality of the scan, as well as the quality of LiDAR cloud points, also verified by overlapping the GCP over the LiDAR points (Table 5).

**Table 5.** Absolut scanning accuracy by verifying LiDAR 3D cloud point over the GCPs for Sânmihaiu Român Lock.

| GCP | GCP/LiDAR | Y (m) | X (m) | Z (m) | DXY | DZ | DXYZ | Horizontal Angle |
|---|---|---|---|---|---|---|---|---|
| GPS1 | GCP | 196,683.443 | 475,180.393 | 86.942 | | | | |
| (Figure 33) | LiDAR | 196,683.469 | 475,180.408 | 86.957 | 0.031 | 0.018 | 0.036 | 59.0495 |
| GPS2 | GCP | 196,678.649 | 475,178.302 | 86.940 | | | | |
| (Figure 34) | LiDAR | 196,678.638 | 475,178.311 | 86.972 | 0.040 | 0.001 | 0.014 | 308.2967 |
| GPS3 | GCP | 196,672.283 | 475,192.938 | 86.942 | | | | |
| | LiDAR | 196,672.284 | 475,192.923 | 86.987 | 0.015 | 0.002 | 0.016 | 175.9638 |
| GPS4 | GCP | 196,677.175 | 475,195.059 | 86.928 | | | | |
| | LiDAR | 196,677.159 | 475,195.080 | 86.911 | 0.025 | −0.022 | 0.033 | 323.2459 |
| GPS5 | GCP | 196,643.789 | 475,221.665 | 85.803 | | | | |
| | LiDAR | 196,643.808 | 475,221.679 | 85.897 | 0.023 | 0.099 | 0.011 | 54.0403 |
| GPS6 | GCP | 196,647.740 | 475,230.543 | 86.384 | | | | |
| | LiDAR | 196,647.749 | 475,230.533 | 86.422 | 0.016 | −0.005 | 0.017 | 136.5286 |
| GPS7 | GCP | 196,633.130 | 475,245.572 | 85.849 | | | | |
| | LiDAR | 196,633.094 | 475,245.601 | 85.954 | 0.045 | −0.011 | 0.046 | 312.0702 |
| GPS8 | GCP | 196,623.064 | 475,258.573 | 86.308 | | | | |
| | LiDAR | 196,623.084 | 475,258.581 | 86.367 | 0.022 | 0.044 | 0.049 | 68.3779 |
| GPS9 | GCP | 196,538.888 | 475,219.742 | 85.520 | | | | |
| | LiDAR | 196,538.833 | 475,219.704 | 85.535 | 0.037 | −0.009 | 0.042 | 259.3324 |
| GPS10 | GCP | 196,632.680 | 475,242.918 | 86.417 | | | | |
| | LiDAR | 196,632.680 | 475,242.919 | 86.435 | 0.002 | 0.015 | 0.015 | 331.5351 |
| GPS11 | GCP | 196,702.977 | 475,298.237 | 86.241 | | | | |
| | LiDAR | 196,703.013 | 475,298.228 | 86.291 | 0.039 | 0.013 | 0.041 | 107.8250 |
| GPS12 | GCP | 196,623.461 | 475,267.737 | 86.615 | | | | |
| | LiDAR | 196,623.471 | 475,267.758 | 86.642 | 0.028 | 0.018 | 0.033 | 25.9656 |
| GPS13 | GCP | 196,621.210 | 475,266.853 | 86.610 | | | | |
| | LiDAR | 196,621.216 | 475,266.861 | 86.636 | 0.020 | 0.007 | 0.021 | 11.9586 |
| GPS14 | GCP | 196,711.086 | 475,213.282 | 84.727 | | | | |
| | LiDAR | 196,711.123 | 475,213.258 | 84.762 | 0.045 | 0.026 | 0.042 | 121.5481 |
| GPS15 | GCP | 196,646.919 | 475,205.019 | 85.713 | | | | |
| | LiDAR | 196,646.915 | 475,205.007 | 85.717 | 0.011 | 0.002 | 0.011 | 206.6933 |
| GPS16 | GCP | 196,641.389 | 475,177.548 | 85.841 | | | | |
| | LiDAR | 196,641.370 | 475,177.554 | 85.865 | 0.015 | 0.023 | 0.028 | 286.0700 |
| GPS17 | GCP | 196,670.360 | 475,188.947 | 85.860 | | | | |
| | LiDAR | 196,670.378 | 475,188.935 | 85.891 | 0.020 | 0.008 | 0.021 | 114.4823 |
| GPS18 | GCP | 196,670.937 | 475,176.200 | 85.849 | | | | |
| | LiDAR | 196,670.996 | 475,176.184 | 85.868 | 0.042 | 0.009 | 0.043 | 104.2688 |
| GPS19 | GCP | 196,612.545 | 475,152.280 | 85.810 | | | | |
| | LiDAR | 196,612.563 | 475,152.278 | 85.851 | 0.018 | 0.033 | 0.043 | 95.2172 |
| GPS20 | GCP | 196,719.979 | 475,193.586 | 84.712 | | | | |
| | LiDAR | 196,719.987 | 475,193.612 | 84.797 | 0.028 | 0.023 | 0.036 | 18.1147 |

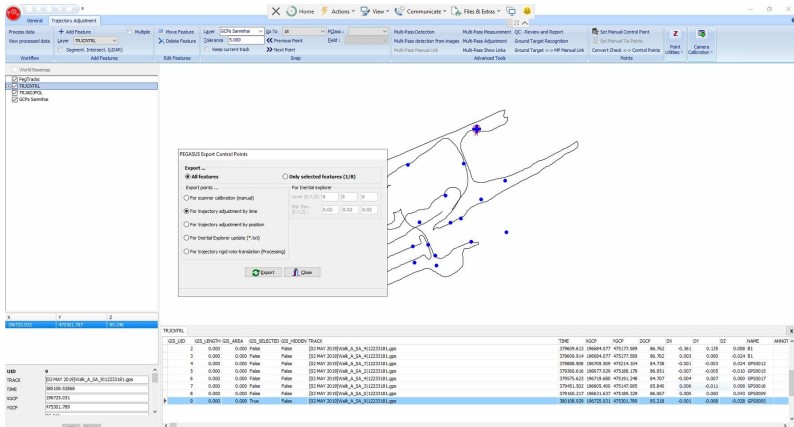

**Figure 34.** Creating a GIS for the Sânmihaiu Roman Lock.

After defining the correct position of the GCPs and positioning them on the point clouds, a GIS will be created that will include, data about the point and from which "Walk" the scan was performed, time, X, Y coordinates and Z of the GCPs selected from the list, the deviations DX, DY, and DZ, as well as the name of the control point (Figure 34).

At the end of the positioning, the data of the new GCPs will be exported and the trajectory will be adjusted according to time (Figures 35 and 36).

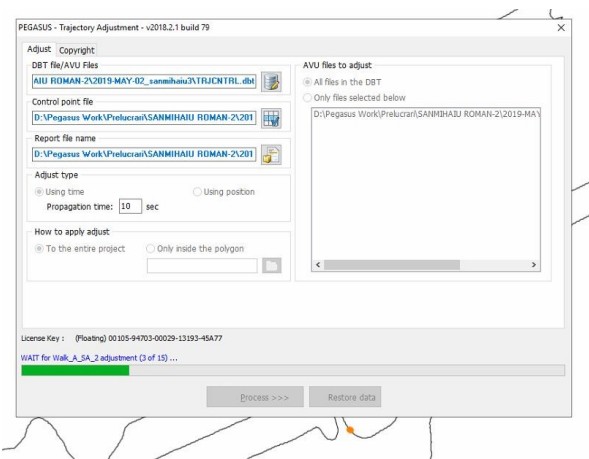

**Figure 35.** Time-adjusted trajectory processing on GCP checkpoints.

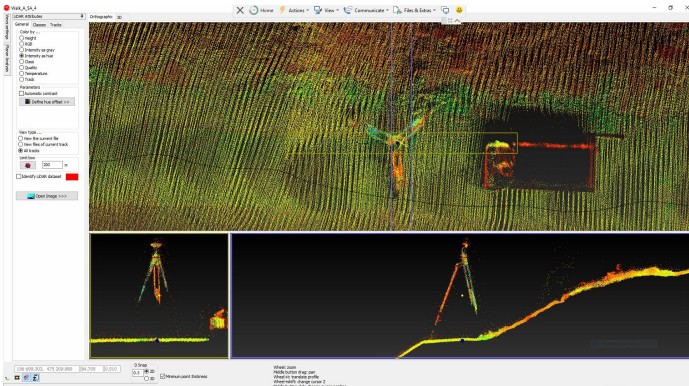

**Figure 36.** Check the adjusted GCPs over the point cloud.

### 4.25. Export of Adjusted 3D Point Clouds

In this last stage, the Pegasus Manager will again export the point-adjusted point clouds (GCP), depending on the time with the LAS extension, E57, an extension that will be processed in the work with the GIS Global Mapper program [51] v.20 (Figures 37 and 38).

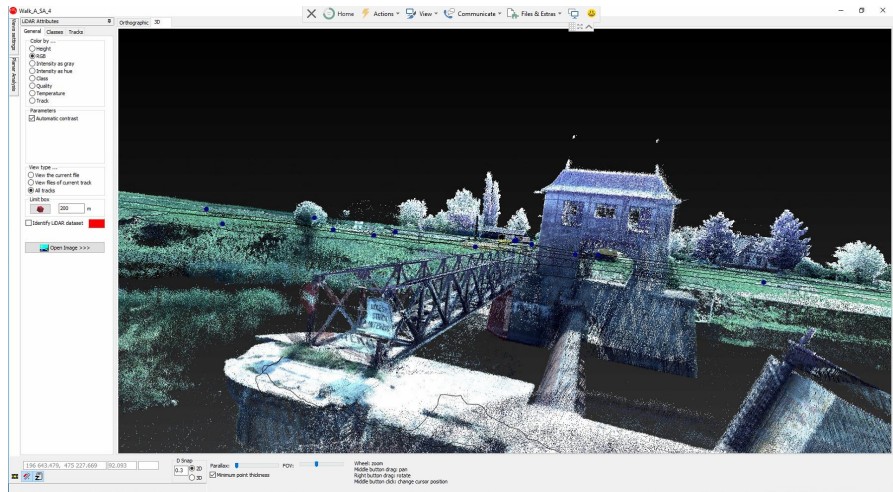

**Figure 37.** View time-adjusted point clouds on GCP.

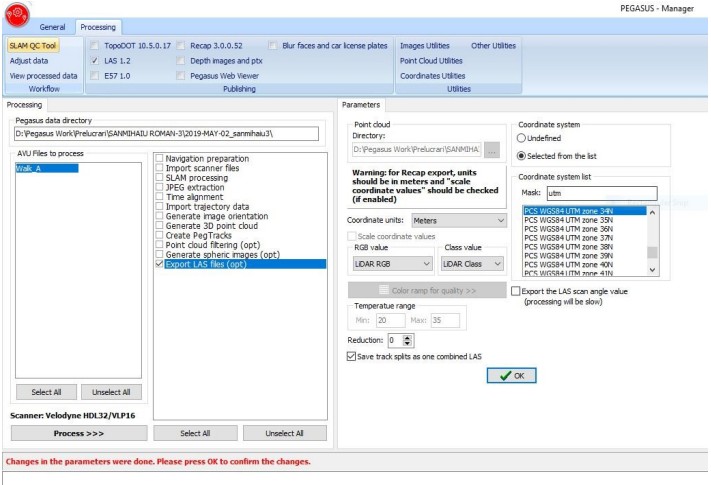

**Figure 38.** Export the LAS file (a file designed for the interchange and archiving of LIDAR point cloud data).

### 4.26. Processing LiDAR Data in ArcGIS–Leica Pegasus: MapFactory

The results obtained from data processing will be further processed with the GIS program, using the MapFactory module, which is a working module in ArcGIS, used from data collection to final extraction.

MapFactory for GIS, MapFactory for AutoCAD, and Cyclone with the CloudWorx module and 3D modeling (Cyclone model) software allows you to view images and LiDAR point clouds to analyze, collect, and export LiDAR data. Once the objects have been measured and meta-labeled, the data can be exported in various formats, including AutoCAD.

Furthermore, the clouds of LiDAR points obtained from the scanning of the Sânmihaiu Român Locks, using the MMS technology with the Leica Pegasus backpack in the Pegasus Manager program, were exported to LAS and E57 LiDAR file that were later imported into Leica's Cyclone Model program where, with the help of the CloudWorx mode for AutoCAD Map and the 3D modeling mode

used, it allows the visualization of digital reality in CAD and the realization of the 3D situation plan. This CloudWorx module uses the AutoCAD work platform and the Cyclone program as a basis for viewing point clouds. Thus, LiDAR points can be viewed, analyzed, vectorized, and exported to CAD.

## 5. Discussion

Measurements using the Mobile Mapping System with Leica's Backpack, the Pegasus Backpack, offers unique three-dimensional (3D) mapping and rapid field reality capture solutions with a navigation system (INS) equipped with global navigation satellite systems (GNSS) and inertial measurement unit (IMU) sensors. In case of natural disasters, the mobile mapping solution (Leica Pegasus) offers real and immediate mapping solutions, with the possibility of representing and extracting topographical elements of the land in the affected areas and with the possibility of exporting LiDAR data in different formats, including LAS or for CAD or architecture programs.

3D laser scanning using LiDAR has become a popular tool today for generating digital terrain models (DTM). Among the latest 3D scanners currently on the market is this model used in research, namely Leica Pegasus Backpack [37] which is a mobile scanning equipment (Mobile Mapping System) that combines a scanning technology with two Velodyne systems [36,40] which can scan both horizontally and vertically, together with five cameras, and where GNSS technology cannot be used, LiDAR data processing will be made using SLAM technology and the Kalman filter.

Backpack use is a valid and cost- and time-effective alternative for DTM extraction, especially if the high ground resolution is required.

These X, Y, and Z measurements, usually photo resisted with high-resolution digital images, can then be viewed, navigated, measured, and analyzed in CAD modeling and design software for feature extraction (e.g., road section), in mapping, and 3D data visualization.

High-speed data collection, huge point density, high survey efficiency and safety are relevant benefits of MMS technology, which is gaining increasing importance in many application areas such as civil engineering, design, road infrastructure, environment, cultural heritage, and others.

Due to continuous developments in both scanning and navigation technologies, the landscape of mobile mapping systems is changing and evolving rapidly. The table below provides a review of the main current trading systems available on the market; in addition, there are several other solutions developed internally at research centers and SMEs (Table 6).

**Table 6.** The most common commercial mobile mapping systems and related components post-processed accuracy values given as RMS.

| Provider | Name | Laser Scanner | | | IMU/GNSS | Digital Camera |
|---|---|---|---|---|---|---|
| | | Sensor (s) | Range | Precision | Pos. Absolute | Resolution |
| TOCON | IP-S3 | 1 scanner | 100 m, @ o100% | 50 mm, @ 10 m (1ρ) | 0.015–0.025 m | Spherical chamber, 8000 × 4000 px |
| TRIMBLE | MX8 | 1–2 VQ-250 | 500 m, @ o80% | 10 mm, @ 50 m (1ρ) | 0.020–0.025 m | Up to 7 rooms, 5 Mpx |
| | | 1–2 VQ-450 | 800 m, @ o80% | 8 mm, @ 50 m (1ρ) | | |
| 3D Laser Mapping | Street Mapper | 1–2 VUX-1HA | 400 m, @ o80% | 5 mm, (1ρ) | 0.050 m | Panoramic camera, 12 Mpx |
| RIEGL | VMX-250 | 2 VQ-250 | 500 m, @ o80% | 10 mm, @ 50 m (1ρ) | 0.020–0.050 m | Up to 6 rooms, 5 Mpx |
| Renishaw | Dynastar S250 | 1–2 scanner (s) | 250 m | 10 mm, @ 50 m (1ρ) | 0.020–0.050 m | - |
| TELEDYNE OPTECH | Lynx SG1 | 2 scanners | 250 m, @ o10% | 5 mm, (1ρ) | 0.050 m | Up to 5 rooms, 5 Mpx and/or panoramic camera |
| | Lynx MG1 | 1 scanner | | | 0.200 m | |
| Leica Geosystems | Leica Pegasus | ZF 9012 | 119 m | 0.9 mm, @ 50 m, o80% (1ρ) | 0.015–0.020 m | 8 rooms, 2000 × 2000 px |
| | | Scan Leica P20 | 120 m, @ o18% | 6 mm, @ 100 m (1ρ) | | |

Moreover, if we keep talking about LiDAR data processing, this was done in the research presented with the help of the Pegasus Manager program, where together with the Inertial Explorer program we have obtained the following:

- Clear data on scan quality; data on scan errors; data on the number of GNSS satellites that were available at the time of the scan; data on areas where we did not have a sufficient number of satellites; and data on time, with the possibility of realigning the time between walks (walks).
- Data on multiple passes through the same place (MultiPass), where point clouds can be colored separately in different colors for each Walk, being able to observe the areas where point clouds double, triple, etc., can be thus take measures to correct point clouds by time or position.
- Viewing areas obstructed by certain elements, or which have registered large errors, where we can accurately observe the position.

In terms of time, this is a very important element during the scans and the SLAM compensation algorithm is the one that indicates the time that will have to be subjected to the compensation processes and the application of this algorithm.

More precisely, "what is the time that will be subject to SLAM compensation", or, how long did we scan without a sufficient number of GNSS satellites (where for each scan or "Walk" we had to have at least five satellites both at the beginning and at the end of each Walk or walk) for various reasons (tall trees, tall buildings, or other obstacles that may disturb visibility to the sky, not having a fully open sky).

LiDAR point clouds obtained from MMS data processing can be RGB colored due to images collected in the field by the five cameras, images that can be collected in the field from meter to meter or the number of images can be adjusted from 1 to 10 images/meter for each camera, adjustment that can only be done when initializing the equipment and creating a new "Job".

It should be mentioned that the Cyclone 9.3 program with the modeling version (Model) has the possibility to make cross-sectional profiles directly from the point clouds and together with the option "Virtual Surveying" and "Smart Pick Point" you can select point by point for the creation of polylines or 3D polygons that can be subsequently exported for processing and making situation plans, in AutoCAD or directly in a file with XML or DXF extension for users of Leica Infinity or CAD programs.

In terms of accuracy and accuracy assessment, the performance test was performed using "reference" data obtained from a photogrammetric survey and a 3D laser scanner (TLS). When comparing point clouds obtained from different instruments, several aspects need to be considered, namely:

Defining a correct range of errors, because each system has its own sources of uncertainty and sensitivity. The 3D points compared do not exactly match each other. For example for the ScanStation Leica C10 the dot clouds resulting from the scan will be aligned with each other and appear as vertical straight lines, while for the Leica Pegasus Backpack, thanks to the 16 laser beams (VLP16) the dot clouds will be placed in shape V, visible in the Cyclone program with which most of the 3D data processing was done.

The surfaces of the object are not evenly digitized because the acquisition positions are different. For these reasons, a methodology is carefully designed to analyze the accuracy obtained by the MMS system with advanced statistical methods.

The precision evaluation of a single one-point cloud highlights the multiple scanning and processing possibilities of the Leica Pegasus mobile system. The estimated errors for the objectives scanned in this paper are characterized by an average dispersion of ±5.7 cm (MMS vs. UAV) and ±1.6–2 cm (MMS vs. TLS).

As expected, errors increase if MMS mapping is performed on narrow streets, with tall buildings and more difficult obstacles. However, the calculated value of ±4.1 cm in such areas is still an acceptable compromise.

To perform scans using the Leica Pegasus backpack for LiDAR data collection, when using a vehicle as a means of transportation, avoid a vehicle speed greater than 20 km/h.

From the research carried out during the elaboration of the paper, an ideal travel speed is 15 km/h. A higher speed will collect less LiDAR data, and viewing and processing point clouds will be more difficult. The higher the speed, the less LiDAR data we will have and the more gaps or missing areas.

Due to the continuous update performed for the Leica Pegasus Backpack, some operating errors were eliminated, and the combinations for the Kalman Filter for guidance, navigation and control decreased from 55 to only 6 combinations, thus reducing the post-processing time.

Another important element was that for the INS, which during one of the scans, where the optimal value for the INS must be ≤ 1, came to increase around 857 m/s (value that increased further), resulting in a speed of 8057 km/h.

## 6. Conclusions

The research topic has a great importance and topicality for the field of hydrotechnics by applying modern techniques for creating a database on heritage conservation and with the possibility of capitalizing on the data obtained whenever needed.

In Romania, the concept of Mobile Mapping is being developed, but for the time being the purchase of these mobile scanning equipment is expensive and requires a rigorous training for this purpose. Socially, the theme is important for the development of the 3D digital database, as well as for the realization of the project on "Revitalization of the Bega Canal" from Timisoara to Zrenjanin (Serbia) until 2021, where Timisoara and Novi Sad will be European Capitals of Culture. In this sense, the works for the restoration of the Sânmihaiu Român Hydrotechnical Node, including the lock, were started, with the term of completion being 2021.

In this sense, several 3D scans were performed for this lock in Sânmihaiu Român, where a first scan was performed before the start of the renovation works, while performing 3D scanning during the rehabilitation works using advanced 3D scanning technologies, following the completion of the renovation to perform an additional scan, in order to be able to the entire database for the conservation and rehabilitation of heritage as needed.

This type of measurements performed using the TLS technique, namely MMS are the first of its kind in Romania, being for the first time when applied for this field, pioneering this way in the field of hydrotechnics in Romania, especially for the Banat Water Basin Administration who manages through this scanning work in order to create a digital data base, which will be used for achieving some projects regarding virtual reality (VR—Virtual Reality) namely achieving BIM (building information modelling) projects [55–59], considering the fact that before starting the rehabilitation works of Sânmihaiu Român Hydro technical Node, two scans were performed using the Leica C10 terrestrial laser scanner as well as a mobile scan using the Leica Pegasus Backpack scanner. There will also be a 3D scan with the Leica Pegasus Backpack mobile scanner at the end of the rehabilitation works, for the revitalization of the Bega River and for tourist navigation, including by capitalizing on tourist locations thus creating unlimited opportunities for development of the area with at least two advantages to complete two European projects already underway: the first project being the one through which the bicycle track was made along the Bega canal, a work in which the authors participated for the topographic surveys on a length of 36 km and the realization of the compensations, respectively the situation plans. The second project, called the Eco Timiș project, is the one that promotes in Timiș the partnership with Serbia as a tourism area. In this context, the 3D scan was performed for the Sânmartinu Maghiar Hydrotechnical Node, located 15 km (from the bicycle track) compared to the Sânmihaiu Român lock.

Lock No.1 Sânmihaiu Român appears in the List of Historical Monuments of Timiş County, drawn up by the Directorate for Culture, under heading 271 with the identification code TM-II-m-B-06283. The official name is The Dam House. The length of the lock room is 70 m and the width is 10 m. It has a 2–3 m fish. There were also promises of rehabilitation in 2008 and 2010, when the canal was cleaned, but today we can also talk about a 3D heritage documentation that can be carried out on the basis of the 3D scales offered by this work.

One of the pursued objectives, within the presented paper, using modern methods of acquisition and processing of topo-geodetic data for hydro technical and hydro-amelioration arrangements is the provision and capitalization of the resulting data, 3D scans by the National Administration "Romanian Waters", The National Agency for Land Improvements, as well as by other interested

agencies/companies, in order to complete the virtual patrimony, made on the basis of modern and ultra-modern equipment, obtained with the help of advanced work programs.

As a comparison in terms of measurements' accuracy, one may notice, that in addition to the savings in staff assigned to a project, it can be said that in the case of performing 3D scans using a mobile scanning instrument will no longer be necessary additional field commissions for completions, where the extraction of the desired elements will be done directly from the office. In this regard, a single 3D scan can be used to carry out several projects.

Regarding the analysis of the accuracy of the measurements obtained with Leica Pegasus, one could notice that the differences between the Leica Pegasus Backpack (MMS) scan and GNSS measurements are about 2–3 cm at an absolute level, and the relative results are even better (in the order of millimeters), 13–20 mm). I noticed this when scanning the Coșteiu, Topolovățu mic or Sânmartinu Maghiar hydrotechnical node, but this equipment was also successfully used to scan the roads and streets from the built-up area of Vinga (Arad County), Oravița (Caraș Severin County), Lipova (Arad County), where for example for the scan of the built-up area of Șeitin locality (Arad County), which consists of 464 streets, with a total length of 36 km, the scan performed from the car took 5.5 h at a maximum speed of the vehicle of 15–18km/h, and by classical measurements with the total station with 9–10 surveyors it would have needed approximately three weeks.

The scanning accuracy based on quality graphics was maximum 18 mm, and by processing the data in the Cyclone program and exporting them to Autocad, according to the performed verifications and compared with the total station, this error was 2–3 cm. However, it has been noticed that at a vehicle speed of more than 20 km/h, point cloud were rarer, which made it difficult to process points for constructions, property boundaries or ditches.

On the other hand, measurements made with the ScanStation Leica C10 equipment require a longer time, with stages similar with a total station, where parking time is different for each station point and automatically calculated by the instrument according to the chosen GRID, and the maximum distance at which the scan is performed, with high accuracies of the order of millimeters, 0–6 mm, while the use of the Leica Pegasus Backpack requires a very short time to perform the scans, and the accuracies obtained are of the order of centimeters, 1–3 cm. Large volume of raw data requires high-performance graphics stations, where, after processing and obtaining point cloud, these final data will be 5–6 times higher than the raw data. Completing the results obtained with the help of modern 3D scanning technologies with photogrammetric results is necessary, where these technologies complement each other and are not excluded.

**Author Contributions:** All authors have contributed to the study and writing of this research; A.Ș. data acquisition and data processing; L.Ș., conceptualization and drew the main conclusions; T.E.M., C.A.P., F.I., T.A., and I.R., analysed the data; R.P., writing—reviewing and editing. All authors have read and agreed to the published version of the manuscript.

**Funding:** The publication of this paper is supported through the project "Assuring Excellence in RDI Activities within the USAMVBT" Code 35PFE.

**Acknowledgments:** This paper is published within the project "Assuring Excellence in RDI Activities within the USAMVBT" Code 35PFE, submitted in Competition Program 1–Development of the National Research and Development System, subprogram 1.2–Institutional Performance, Institutional Development Projects–Excellence Funding Projects in RDI.

**Conflicts of Interest:** The authors declare no conflict of interest.

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
