# Peer review of "Use of Modern Technologies for the Conservation of Historical Heritage in Water Management"

_water, doi:10.3390/w12102895_

Round 1

Reviewer 1 Report

The manuscript presents the use of combined surveying technologies, such as MMS, TLS, aerial photogrammetry, and GNSS georeferencing, applied to a cultural heritage case of study: the Sânmihaiu Român Sluice located in TimiÈ™ County (Romania).

The work reports mainly the data acquisition and post-processing through a Leica Pegasus Backpack mobile scanning equipment and related software. The workflow is very detailed, resembling in some parts more a report than a research paper.

The reviewer has noticed a disequilibrium between the methodology and instrument description and the main purpose of the paper, i.e. describing how this technology can be usefully applied for cultural heritage preservation and conservation in relation also to the water environment.

Furthermore, the overall structure of the paper is confusing and not clear in some parts. Some suggestions will be given by the reviewer to improve the quality of the paper:

  • The abstract does not clearly present the main purpose of the work. The used technology should only a tool for providing useful data for the preservation and conservation of the selected study case. The words “complete, completion” can mislead the final aim of the research.
  • The introduction contents are not well linked between the paragraph, resulting quite confusing. The suggestion is to review the contents and make them more connected and fluid; here some indications to improve legibility:
    • Some phrases are too long, as in 44-56, presenting the topic with too general considerations.
    • From line 57 to line 62 and line 75-80, it seems advertising of the instrument, not just a description, thus being not appropriate for a research paper.
    • Line 63, the subject is missing
  • The Materials and Methods section is not very balanced towards the technical content.
    • In fact, from lines 132 to 200 almost a historical background of the study case is reported. The suggestion is to reduce this part and to move to a separate paragraph dedicated to the description of the Sânmihaiu Român hydrological node history and importance under the hydrological and cultural point of view. Moreover, also the historical sources must be cited (“Historians claim that…” is not enough).
    • The technical part reports the use of a DJI (please, specify the builder) Phantom 4 Pro UAV for aerial photogrammetry. But, nor the term “photogrammetry” nor the results are reported in the paper. A comparison with the point clouds obtained by MMs and TLS would be interesting. In lines 267-271 the statements about SfM compared to other sensors are not supported by experimental results or citations of other research works.
    • Line 252, please describe better how the errors were evaluated
    • Lines 360-387 the description of the Leica backpack is repeated exactly as in the previous page. Please, check redundancies over the entire manuscript
  • The Results section presents a very detailed workflow for what concerns data acquisition and processing: some of this information may be ascribed more to the material and Methods sections. This form also reminds a report more than a research innovative content, with excessive attention to the software and instrument specifics and procedures. The suggestion is to reduce these workflow explanations to the very essential (maybe detail them in supporting material contents), please also check the figures captions and if all of them are necessary. This section would be useful to present the results of the different techniques used and compare their outputs (point clouds) to verify their complementary and to highlight the qualities and weaknesses of each.
  • The Discussion section sums up all the consideration based on the obtained outputs:
    • Lines 967-972 seems a repetition of the instrument description reported elsewhere

Finally, please double check typos and grammar errors over the entire manuscript. Some English style corrections should be made, thus a review from a mother tongue is strongly suggested.

Author Response

Dear Reviewer, 

Thank you very much for your specific observations and remarks in order to improve our manuscript. They were really helpful in reshaping all the manuscript.

Thus, we have modified everything according to your suggestions.

Please find enclosed, point by point, all the responses.

Thank you, once again for your support and cooperation!

Best regards!

Reviewer 2 Report

rows 57-63 why is the text in Bold?

106 without GNSS / INS sensors,...no, it was more times written, that is GNSS / IMU or complet INS

117 Of course, IMMS...MMS?

176 the Second World War...World War II

235  minimum of 5 GPS satellites, optimally...please, fing all GPS abbreviation in the text and change it on GNSS, it was written in firs review; you still use once gps, once gnss...improve it

240 Bei Du... BeiDou

280-284, 324-327 why is the text in italic?

337 The stable relative position of the INS can be used to go through the times when the GNSS solution is degraded or unavailable....you mean IMU; it can be used in times, when a GNSS signal fail

362 GNSS/INS ...still wrong  - GNSS / IMU

377 in single inertia  ...inertial

382 first, GNSS and INS data are integrated...still wrong :-(

396 why is the text in bold?

401 ALMANAC Data  ... data; on 408 almanac data?

411 DINAMIC initialization...dynamic

477-483 RiNEX vs. RINEX

501 scanned lenses? What do you mean? lenses

540 improve GNSS and INS quality..still the same error

592 Standard deviation: >0.05;  metres?

Many images only show different photos or images of objects or different colored point clouds. It is unnecessary. The article then looks like a textbook.

Much of the text is about GCPs. There are unnecessary pictures showing GCP. But what were they used for? Where are any deviations from GCP and self-measurement?

961 networks. .

970 are accompanied by GNSS, INS and IMU...???

Conclusion:

In a conclusion, results of experiments, mesurements etc. should be discussed. There arent experiments, measuremens, which show precision of the Pegasus system based on independent other measurement. In this paper, only using the Pegasus measurement is shown. On row1062 This project represents a real support for water management... With what? Will we use scanning to improve the condition?

1077 bold text?

How you will make the data visualisation? The documentation should used for a BIM in your project...see for example see "Laser scanning for BIM and results visualization using VR, ISPRS " or "Combined 3D building surveying techniques-Terrestrial laser scanning (TLS) and total station surveying for BIM data management purposes". Complete the literature, there is a lot of company materials.

Sorry, a lot of the text looks like an instruction manual, it is unnecessary in a scientific article; also there are too many pictures (66) and the article is too long also due to the coordinate tables (irrelevant if they do not show any differences in measurements by different methods).

If you want to publish this article, ok, it was a hard work, but shorten it, take out unnecessary pictures and finally make some comparison of technologies in terms of accuracy and speed, e.g. or analysis of the accuracy of the Pegasus measurements versus other measurements (GNSS, total station, etc.).

Author Response

Dear Reviewer, 

We all thank you for your suggestions and remarks, which were very useful in reshaping the entire manuscript.

Consequently, we have modified the entire document, and really appreciated the accuracy of all your observations, and responded to all of them, step by step.

We hope that finally we have accomplished a good article. 

Please find enclosed the responses to all your requests.

Thank you once again for your support and cooperation!

Best regards!

Round 2

Reviewer 1 Report

 The reviewer is happy with the revisions made by the authors as per the given suggestions.

Author Response

Dear Reviewer,

Thank you so much for your support and cooperation, but above all, for your guidance along the entire process of evaluation.

Your precise observations and guidance helped us to reshape the manuscript and to turn it into a more fluid one and a more coherent one, with a certain clear line.

We are grateful for your patience and efforts in guiding us towards the current form of our manuscript!

We hope to have done a good work and we really appreciate your efforts.

Thank you!

Best regards!

Reviewer 2 Report

Sorry, I find still some problems or inaccuracy, but insignificant. You can easily fix it.

97 SFM should be... SfM

99 ok, I find it good, some words about history

I think, there are  many abbreviations; some are notorious known and do not need to be explained (laser, UAV), some need to be described at the first occurrence, i.e. best in Introduction; but somewhere on the paper beginning (of course, not in abstract) and one times only; next, you can use abbreviations only.

for example:

208 GCP (Ground Control Point), next on 232 should 12 GCP´s only, 237, 265, 274, 736 etc. find next...

214 SLAM (Simultaneous Localization And Mapping), next on 369 etc

251 terrestrial scanning (TLS) should be terrestrial laser scanning

286  they provide liDAR 3D point clouds (LiDAR Point Cloud)...should be LiDAR, but I thing ScanStation and other produce a typical point cloud, why liDAR 3D point clouds (LiDAR point cloud)?

297 you use IMU, but abbreviation explaining is on 305, 336, 340, 352 etc

321 ZUPT Zero Velocity Update Algorithm), 386; but on 392 is ZUPT Zero Velocity Update, 428

324 INS (inertial navigation system), next on 351 etc

721, 734 LiDAR or LIDAR, joint it to LiDAR

877 After defining the correct position of the GCP ground control points...leave only GCP, 885 GCP control points... you write actually: ground control point control point-that does not make sense, GCP is a control point

913   5. Discussion  ... abbreviations cannot be explained until the end of the text (LiDAR is on row 300, but it was used in the text before abbreviation  explaining), 920 SLAM (Simultaneous Localization and Mapping), it is on 214, 369

914   3D laser scanning (called LiDAR)...no, LiDAR is an instrument; 3D laser scanning (using LiDAR...). I thing as LiDAR is offten called laser scanner on mobile carrier, mainly on an airplane; but on a car or backpack to. For a typical terrestrial laser scanner (for example Leica ScanStation) the TLS abbreviation can be used

915 DEM ...digital elevation model: it is a regular grid of 3D point; TLS or ALS or MMS generate DRM (digital relief model, may by, you can use DTM - digital terrain model as you use in row 921

389 OEKF, it was explained on 324

1018 the Leica C10 terrestrial laser scanner ...(Terrestrial Laser Scanning)..it is unnecessary, when you write about the Leica C10 terrestrial laser scanner

***

The article is already significantly better, only problems with technical terms and abbreviations remain. After correction, it can be recommended for publication.

Author Response

Dear Reviewer,

We have followed all the indications from your review on our manuscript, and according to your guidance, we have reshaped the manuscript, taking out the explanations of all the abbreviations, so that we have their explanations only once in the entire text. We have also found other additional mistakes with other abbreviations, except the mentioned ones, and we have corrected them accordingly.

We have carefully respected and remade each point from your appreciation.

We really hope to have completed all the tasks and to be able to give another shape to our manuscript.

We do thank you for your appreciation, observations and guidance.

We really see now the manuscript more fluid and more coherent and clear.

Thank you for your support and cooperation!

Best regards!

This manuscript is a resubmission of an earlier submission. The following is a list of the peer review reports and author responses from that submission.

Round 1

Reviewer 1 Report

There are still some failings:

row 45  Yugoslav territory...Yugoslavia dont exists more.

79  the EU currency is EUR (ISO 4217)

124 GNSS and IMU sensors (not INS)

143 GNSS and INS data...false, INS= GNSS + IMU; it was in first review more times cited

150 combines precision GNSS receptors with robust inertial measurement units (INS)  - (IMU!!!)

159 provide a combined GNSS + INS   ...no, GNSS+IMU!!!

161 attitude  , no, altitude

165 An inertial navigation system (INS) is a system that is used to calculate the relative position over time from the rotation and acceleration information provided by an inertial measurement unit (IMU).   Wrong, it is IMU; this device measures the relative position over time from the rotation and acceleration information. Calculated relative position is controlled by GNSS

168 When an INS is combined with GNSS... one more, INS=GNSS+IMU; please, look at the theory, this is a mix

179 The navigation system (INS) is equipped with GNSS and IMU sensors, with absolute altitude and outdoor positioning by GNSS...ok, but this info is more time repeated, fix it

214, 241 etc. How long was it scanned without a GNSS signal or less than 5 GPS satellites? GNSS or GPS?   GPS is an American system only

317 etc RiNEX or RINEX?

355 Camera calibration .. without description these tables they don't matter. Describe it. It is by you in the project made calibration (you did it yourself?)

562 The system uses only GPS Navstars  satellites?

630 GCPs

It will be good some sentences about the accuracy of system. I think, you need necessary control points and on these control points you get after processing discrepancies between GCPs (signalized and measured directly with a GNSS aparature) and measured points.

References

add more references on MMS, there are too much references to firm material. You can find a lot of papers on the web (e.g. A Survey of Mobile Laser Scanning Applications and Key Techniques over Urban Areas or Using mobile laser scanning data for automated extraction of road markings or  Automatic classification of point clouds for highway documentation)

Do you read some articles about UAV and MLS, TLS? (e.g. Comparing terrestrial laser scanning and unmanned aerial vehicle structure from motion to assess top of canopy structure in tropical forests) What is the advantage of MLS with Pegasus comparing other techniques???

Reviewer 2 Report

Overall

The manuscript is more like a "Technical Note" and not "research paper" where the benefits of Mobile Mapping System (MMS) are described. The intention of the authors to describe the survey by this technology is clear but poorly represented. The authors should show the accuracy achieved with this MMS on control points and compare the results obtained with other sensors/methods/technology. In addition, not even a picture of the point cloud is shown in the manuscript.

The abstract must be completely rewritten. In this part of the manuscript, it is necessary to summarize the results obtained and the fields of application to which the paper is addressed.

Everything written in the introduction must be moved into the experimental part, i.e. in the Materials and Methods section. In the introduction, it is necessary to write how and where your work is placed in the scientific context. Therefore, in this part the aspects of geomatics survey must be highlighted.

The section called Discussion is rather poor in content and does not add quality to the paper.

In the references, a double numbering has been used for the same paper (10 and 23).

Finally, a bold or italic typeface was often used in the manuscript, which is almost always to be avoided unless strictly necessary.

Details

Line 36            write better the unit of measurement km2

Line 177          This is not the equipment but the box containing (probably) a sensor.

Line 206          Specify the acronym GPS

Line 208          Specify the acronym GNSS

Line 235          The picture must be modified and made with more advanced tools, such as ArcMap software, where are reported the metric scale and north. In addition, the labels must be reported in an easily readable font.

Line 215          I suggest deleting this sentence …

Line 250          I suggest unifying the figure 3 and 4 describing the two steps

Line 330          Specify the acronym RTK

Line 355          I suggest to describe the calibrations operations and to move the tables in a specific Appendix

Line 572          Replace the word "air" with "airborne"

Line 595          Specify the acronym BIM

Reviewer 3 Report

The english language and grammar of the manuscript require significant edits.

There is no clear new and original content. The paper reads like a detailed user manual for the Leica backpack system including screenshots.  This may be useful for someone trying to use the system, but the topic is not appropriate for a peer reviewed journal article.  There is no novelty or scientific insight gained from the paper beyond describing a demo project that the authors did with a commercial hardware and software system.